# Do Vision-Language Models Represent Space and How? Evaluating Spatial Frame of Reference under Ambiguities

**Zheyuan Zhang**[1][*]    **Fengyuan Hu**[1][*]    **Jayjun Lee**[1][*]
**Freda Shi**[2,3]    **Parisa Kordjamshidi**[4]    **Joyce Chai**[1]    **Ziqiao Ma**[1]
[1]University of Michigan  [2]University of Waterloo  [3]Vector Institute  [4]Michigan State University
https://spatial-comfort.github.io/

## Abstract

Spatial expressions in situated communication can be ambiguous, as their meanings vary depending on the frames of reference (FoR) adopted by speakers and listeners. While spatial language understanding and reasoning by vision-language models (VLMs) have gained increasing attention, potential ambiguities in these models are still under-explored. To address this issue, we present the COnsistent Multilingual Frame Of Reference Test (COMFORT), an evaluation protocol to systematically assess the spatial reasoning capabilities of VLMs. We evaluate nine state-of-the-art VLMs using COMFORT. Despite showing some alignment with English conventions in resolving ambiguities, our experiments reveal significant shortcomings of VLMs: notably, the models (1) exhibit poor robustness and consistency, (2) lack the flexibility to accommodate multiple FoRs, and (3) fail to adhere to language-specific or culture-specific conventions in cross-lingual tests, as English tends to dominate other languages. With a growing effort to align vision-language models with human cognitive intuitions, we call for more attention to the ambiguous nature and cross-cultural diversity of spatial reasoning.

## 1 Introduction

The recent success of large language models has sparked breakthroughs in multi-modalities, leading to the development of many vision-language models (VLMs; Chen et al., 2023c; OpenAI, 2024; Reid et al., 2024, *inter alia*). With some benchmarks developed to evaluate the downstream performance of these models (Liu et al., 2023c; Yue et al., 2024), there has been growing excitement around evaluations and analyses inspired by human cognitive capabilities such as referential grounding (Ma et al., 2023a), compositional reasoning (Ma et al., 2023c), visual illusions (Zhang et al., 2023; Guan et al., 2024), and theory of mind (Jin et al., 2024). One direction among them that captures significant attention is spatial language understanding and reasoning, leading to several benchmarks (Mirzaee et al., 2021; Mirzaee & Kordjamshidi, 2022; Kamath et al., 2023; Liu et al., 2023a) and enhanced models (Chen et al., 2024a; Cheng et al., 2024; Premsri & Kordjamshidi, 2025).

Indeed, spatial cognition is a crucial part of human cognitive capability developed in early ages (Tommasi & Laeng, 2012; Vasilyeva & Lourenco, 2012). Language is closely intertwined with spatial cognition, with each contributing to the acquisition of the other (Hayward & Tarr, 1995; Regier & Carlson, 2001; Pyers et al., 2010; Pruden et al., 2011; Gentner et al., 2013). While spatial language and non-linguistic spatial representations in memory are closely correlated and share foundational properties, they are, to some extent, divergent—spatial conventions are not consistently preserved across different languages or tasks, and humans demonstrate flexibility in using multiple coordinate systems for both non-linguistic reasoning and linguistic expressions (Munnich et al., 2001; Shusterman & Li, 2016). Thus, spatial language is inherently ambiguous.

In situated communication, even a simple spatial expression like "the basketball to the right of the car" may have multiple interpretations. People may use different *frames of reference* (FoR; Levinson, 1996; Frank, 1998, *inter alia*) to resolve ambiguity about the underlying coordinate system, as illustrated in Figure 1a. The diversity of conventions across languages and cultures further

---

[*]Authors contributed equally to this work.

**(a) Frame of Reference (FoR)**

Is the basketball to the right of the car?
- Yes, from the camera's viewpoint
- Yes, from the woman's viewpoint
- Yes, from the car's viewpoint

**(b) Coordinate Transformation**

The ball to the left/right/
front/back of the blue ball.
- Translated: A/B/C/D
- Rotated: B/A/D/C
- Reflected: A/B/D/C

E.g., *Hausa*   E.g., *Tamil*   E.g., *English*

**(c) Spatial Continuity**

Is the red ball to the
right of the blue ball?

Figure 1: In situated communication, spatial language understanding and reasoning are often ambiguous, leading to varying interpretations among people from different cultural backgrounds. Specifically: (a) different frames of reference can result in different interpretations of the same spatial expression; (b) speakers of different languages may use distinct coordinate frames for non-fronted reference objects; and (c) spatial relations extend beyond exact axes to include acceptable regions.

| Origin | Frame of Reference | Example (English) |
|--------|--------------------|--------------------|
| Camera (Preferred) | Egocentric Relative FoR | (From the camera's viewpoint,) the ball is **behind** the car. |
| Addressee | Addressee-Centered Relative FoR | (From the woman's viewpoint,) the ball is to the **left** of the car. |
| Reference | Object-Centered Intrinsic FoR | (From the car's viewpoint,) the ball is to the **right** of the car. |

Figure 2: An illustrative example of how a frame of reference (FoR) specifies the reference system when describing the spatial relation between a target object (i.e., the ball) and a reference object (i.e., the car). When the FoR is not explicitly specified, English prefers an egocentric relative FoR, i.e., "the ball is behind the car." We study FoRs that lead to ambiguity (Liu et al., 2010).

complicates this ambiguity—different languages employ different conventions in choosing one FoR among multiple competing options. As shown in Figure 1b, speakers may project themselves onto the ball or consider an imaginary listener facing them (Shusterman & Li, 2016). These ambiguities are not easily resolvable based solely on linguistic expressions (Tenbrink, 2004; Liu et al., 2010).

Our main research question is not new: *Do vision-language models represent space, and how?* Several benchmarks (Kamath et al., 2023; Liu et al., 2023a) have been developed for this purpose, consisting of text-image pairs where objects may or may not follow certain spatial relations. However, the aforementioned spatial ambiguities remain largely under-explored when studying VLM-based spatial language understanding and reasoning. We emphasize that FoRs are crucial to studying spatial cognition across modalities, as they provide a foundational framework for understanding how spatial relationships are perceived, interpreted, and communicated (Levinson, 2003).

To fill this gap, we present COnsistent Multilingual Frame Of Reference Test (COMFORT), a framework that systematically evaluates the spatial reasoning capabilities of VLMs, emphasizing consistency in understanding ambiguous and disambiguated spatial expressions. COMFORT introduces (1) a set of spatial reasoning tasks instantiated by synthetic 3D images and corresponding text describing spatial relations and (2) metrics to evaluate the robustness and consistency of the model responses. We extend the setup to multilingual settings by evaluating models in 109 languages across 170 regions worldwide. We find that VLMs show alignment with English conventions in spatial language understanding when resolving ambiguities. However, they (1) are still far from achieving robustness and consistency, (2) lack the flexibility to accommodate multiple FoRs, and (3) fail to adhere to linguistic and cultural conventions in cross-lingual tests, as English tends to dominate other languages. With a growing effort to align vision-language models with human cognition, we highlight the ambiguous nature of spatial language and call for increased attention to cross-cultural diversity in spatial reasoning.

## 2 BACKGROUND AND RELATED WORK

### 2.1 SPATIAL LANGUAGE AND SPATIAL REPRESENTATION

Some projective terms, such as the English words *front*, *back*, *right*, and *left*, convey meanings of spatial relations (Eschenbach, 2004). These terms articulate the spatial relation between two entities within a designated *frame of reference* (FoR), often involving one entity as the reference object (*relatum/ground*) and another target object (*referent/figure*) that is positioned relative to the relatum

along a specific axis/direction (Levinson, 1996; Frank, 1998). In situated communication, speech act participants (e.g., an *addressee*) may also be considered (Danziger, 2010). To determine acceptable uses of various spatial relations, existing theories suggest that people fit *spatial templates*, which are centered on the relatum and aligned with the FoR (Logan & Sadler, 1996), to parse out *regions of acceptability* of certain directions (Franklin et al., 1995; Carlson-Radvansky & Logan, 1997).

**Ambiguities in frame of reference.** The choice of perspectives may lead to different FoRs, where Levinson (2003) has identified three main types of FoR: *absolute*, *intrinsic*, and *relative*. The absolute FoR uses cardinal directions, such as *north* and *south*, as fixed bearings. The intrinsic FoR aligns the origin with the relatum, describing the referent's position relative to the relatum's inherent orientation. The relative FoR positions a *viewer* (egocentric or addressee) as the origin, focusing on the observer's intrinsic perspective. Liu et al. (2010) have highlighted the ambiguities in situated communication among three variations of intrinsic and relative FoRs (Figure 2): the *egocentric relative*, the *addressee-centered relative*, and the *object-centered intrinsic* FoRs.[1] When not specified, these FoRs are not easily distinguishable based solely on their linguistic expressions (Tenbrink, 2004). To resolve the ambiguity, individuals from diverse linguistic and cultural backgrounds adopt different preferences and conventions in choosing FoRs (Majid et al., 2004; O'Meara & Báez, 2011; Bohnemeyer et al., 2014; Bender et al., 2020; Ogelo & Bylund, 2024).

**Ambiguities in relative FoRs.** The variations of relative FoRs form another source of ambiguity. After putting the origin of the coordination system on the viewer, multiple strategies specifying how to transform the axes can be considered (Figure 1b). Different languages use different transformation conventions to resolve the ambiguity of the front-back and left-right of a relatum (Levinson, 2003; Shusterman & Li, 2016), including: (1) *translated* projection (e.g., Hausa) where the coordinate frame of the speaker is directly applied, (2) *rotated* projection (e.g., Tamil), where the coordinate frame of the speaker is transformed with a 180-degree rotation, and (3) *reflected* projection (e.g., English), where only the front-back axis is reversed.

## 2.2 Spatial Understanding in Vision-Language Models

Large language models (LLMs) have exhibited strong adaptability that extends beyond text, encompassing 2D and 3D vision (Tsimpoukelli et al., 2021; Alayrac et al., 2022; Yang et al., 2024), their affordances in the physical embodiment (Driess et al., 2023; Qian et al., 2024), and various other modalities (Yu et al., 2024a). Especially, a variety of vision-language models (VLM) have been developed by visual instruction tuning on paired text-image data (Dai et al., 2023; Liu et al., 2023b; Dong et al., 2024). With supervised fine-tuning using entity-phrase mappings in text-image pairs, grounded VLMs have been developed for fine-grained vision-language understanding at both the region (Chen et al., 2023a; Bai et al., 2023; You et al., 2023; Peng et al., 2024) and pixel level (Lai et al., 2024; Xia et al., 2024; Rasheed et al., 2024; Zhang et al., 2024).

Spatial understanding is known to be challenging even for state-of-the-art VLMs and is receiving increasing attention (Achiam et al., 2023). In addition to using explicit spatial language understanding modules (Rajabi & Kosecka, 2024), recent works such as SpatialVLM (Chen et al., 2024a) and SpatialRGPT (Cheng et al., 2024) improve spatial reasoning in VLMs by leveraging 3D VQA or scene graph data for supervised fine-tuning. Several benchmarks have also been developed to evaluate spatial reasoning in VLMs from various perspectives (Liu et al., 2023a; Cheng et al., 2024; Kamath et al., 2023). Still, these benchmarks overlook ambiguities related to the FoR, lack spatial continuity, and have not proposed metrics to evaluate the robustness and consistency of spatial reasoning.

## 3 Consistent Multilingual Frame of Reference Test (COMFORT)

We introduce the COnsistent Multilingual Frame Of Reference Test (COMFORT), a new evaluation protocol with dataset, tasks, and comprehensive metrics, to study VLM behaviors in spatial language reasoning with FoR-related ambiguity. This protocol accommodates spatial continuity and various ambiguities, drawing insights from several well-defined metrics to assess performance and prediction consistency. Given our primary focus on analytical and scientific inquiry rather than competitive testing only (Warstadt & Bowman, 2022; Saxon et al., 2024), in this work, we aim to develop better performance and consistency metrics to deepen our understanding of model capabilities.

---

[1]We exclude the absolute FoR from our study as it introduces little ambiguity (Liu et al., 2010).

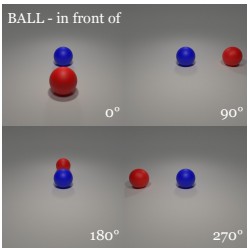 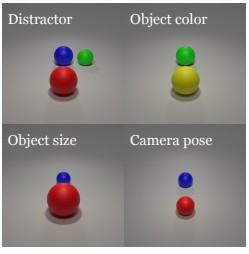  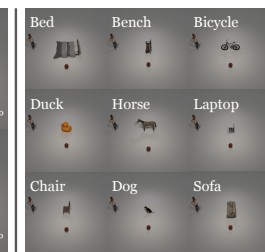

(a) Sample images from `COMFORT-BALL`. The 4 images on the left are selected every 90° interval along the rotational path out of 36 images. The 4 images on the right illustrate variations with a distractor, different object colors, sizes, or camera poses.

(b) Sample images from `COMFORT-CAR`. The 4 images on the left are selected every 90° interval along the rotational path out of 36 images. The 9 images on the right are sample images of each variation with different relatum objects.

Figure 3: Examples from the `COMFORT-BALL` and `COMFORT-CAR` datasets.

## 3.1 TASK FORMULATION

Following the setups in object hallucination evaluation (Li et al., 2023; Chen et al., 2024c), we formulate the task as a spatial relation inference problem. In this task, a VLM $\mathcal{M}$ is presented with an RGB image $x_{\text{img}}$ and a textual question $x_{\text{query}}$. The image shows the egocentric perception of a scene $s \in \mathcal{S}$, where $\mathcal{S}$ is the set of possible scenes in which the referent moves along a rotational trajectory with a constant radius from the relatum. In contrast to fixing the referents on the standard canonical axes, this setup better mirrors the spatial continuity in common real-world scenarios. A language prompt (Table 1) queries whether a spatial relation $r \in \mathcal{R}$ is satisfied by a referent-relatum pair in the image under FoR $f \in \mathcal{F}$ (Figure 2) in language $\ell \in \mathcal{L}$. This work also examines models using queries with no FoR specified; therefore, a test case in COMFORT is defined as a 4-tuple in $\mathcal{S} \times \mathcal{R} \times (\mathcal{F} \cup \{\emptyset\}) \times \mathcal{L}$. While there are many spatial relations in daily languages, we primarily focus on four canonical directions; that is, the considered relation set $\mathcal{R} = \{\textit{to the left of}, \textit{to the right of}, \textit{in front of}, \textit{behind}\}$. COMFORT covers $|\mathcal{L}| = 109$ languages worldwide; however, we use English as an example to describe the data synthesis and evaluation processes for simplicity and clarity, and refer readers to Appendix A for more details.

## 3.2 SCENE SETUP

We render the scenes into images using Blender (Blender Online Community, 2016). Each scene consists of a referent and a relatum. The referent follows a rotational trajectory with a constant radius from the relatum to implement spatial continuity. Starting from the canonical front direction, we move the referent with a uniform step of $10°$, totaling up to 36 images per scene. In COMFORT, there are configurations determined by whether the relatum has an intrinsic semantic front:

- **COMFORT-BALL**: When the relatum is non-fronted (e.g., Figure 1b), we focus on the ambiguity of FoR conventions associated with different languages. The split involves an observer's egocentric perception of a referent (e.g., a red ball) and a non-fronted relatum (e.g., a blue ball). We further randomize the dataset with object-level (colors, sizes, and shapes) and scene-level variations (camera positions and distractors) to consider more diverse yet reasonable settings (Figure 3a).
- **COMFORT-CAR**: When the relatum is fronted (e.g., Figure 1a), multiple FoRs can be explicitly adopted to interpret the scene. A COMFORT-CAR image, therefore, involves the egocentric perception of a referent, a fronted relatum, and an additional human addressee. One can interpret the spatial relations using either the Camera, Addressee, or Relatum (C/A/R) as the origin to resolve the reference frame ambiguity. COMFORT-CAR has a set of 10 realistic objects in a typical household or outdoor scene, including *horse*, *car*, *bench*, *laptop*, *rubber duck*, *chair*, *dog*, *sofa*, *bed*, and *bicycle*, all of which have a clear semantic front. We use a basketball as the referent and vary the relatum. In addition to these objects, we include a human addressee in the scene. To disentangle different FoRs as much as possible, we let the addressee face right, and let the relatum face either left or right in the rendered images from the rendering camera's perspective (Figure 3b).

## 3.3 LANGUAGE QUERY SETUP

Given a pair of referent [A] and a relatum [B], together with a spatial relation of interest, the query is posed as "Is [A] [relation] [B]?" Depending on whether or not and which FoR is specified, the

| Origin | Prompt Template |
|--------|-----------------|
| nop | Is [A] [relation] [B]? |
| cam | From the camera's viewpoint, is [A] [relation] [B]? |
| add | From the [addressee]'s viewpoint, is [A] [relation] [B]? |
| rel | From the [relatum]'s viewpoint, is [A] [relation] [B]? |

Table 1: The origins of each coordinate system and the corresponding prompt templates for querying the FoR given a referent-relation-relatum triple.

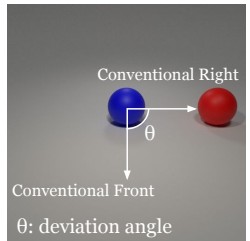

Figure 4: A red ball with a deviation angle $\theta = 90°$ relative to the conventional front (English) of the blue ball.

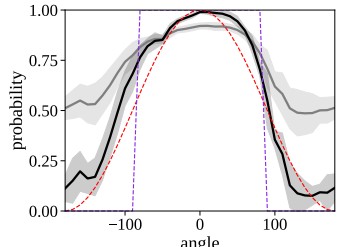

Figure 5: The raw probability $p(\theta)$ in gray, normalized probability $\widehat{p}(\theta)$ in black, and two reference probability $\lambda^{\text{hemi}}(\theta)$ and $\lambda^{\text{cos}}(\theta)$ in red and purple.

query is appended after four different perspective prompts (Table 1): no perspective (nop), camera perspective (cam), addressee perspective (add), and relatum perspective (rel). We only query from the camera egocentric perspective (cam) for COMFORT-BALL, focusing on the ambiguity introduced by variations of the relative FoR. For COMFORT-CAR, we use all four possible language prompts to study how ambiguity in the reference system is resolved. Overall, for English, the above data generation pipeline leads to 720 test cases in COMFORT-BALL, and 57.6k test cases in COMFORT-CAR. The same method for dataset synthesis can be generalized to any other language; however, for computational efficiency, we only include the scenes corresponding to the four most prototypical directions (i.e., left, right, front, and back) in our multilingual analysis.

## 3.4 PERFORMANCE METRICS

Quantitatively assessing the spatial understanding and reasoning capabilities of models is challenging for two reasons. First, the physical world is continuous, and spatial relations may extend beyond the precise canonical front-back and left-right axes. As noted by Carlson-Radvansky & Logan (1997), there exists *regions of acceptability* where, for instance, an object slightly to the front-left might still be considered being "in front." Second, as Dentella et al. (2023) pointed out, language models are biased towards affirmative responses. However, the intermediate representations may be sensitive to variations in input and, to some extent, align with human perceptions of spatial cues. Based on these concerns and findings, we introduce multiple performance metrics, in addition to the vanilla accuracy, to enable more nuanced analyses.

Unless further clarified, we adopt a right-handed coordinate system with the thumb pointing upwards when describing angles. We define the *deviation angle* $\theta \in (-180°, 180°]$ as the angular displacement from the canonical direction $r$ to the vector connecting the relatum and target. For example, in Figure 4, the deviation angle of canonical right from canonical front is $\theta = 90°$. Following Carlson-Radvansky & Logan (1997), we define the acceptable region for a spatial relation $r$ as the 180-degree hemisphere centered at the corresponding canonical direction. For a VLM $\mathcal{M}$ and a test case indexed by $i$, we let $P_i(\text{response}; \mathcal{M})$ denote the probability of $\text{response} \in \{\text{Yes}, \text{No}\}$ assigned by $\mathcal{M}$, and abbreviate it as $P_i(\text{response})$ if there is no confusion.

**Accuracy.** Given a spatial relation $r$ in the textual prompt, we assess whether the assigned response probabilities correspond to whether the referent lies within the acceptable region defined by the relatum and $r$. Formally, we define the local probability of the model responding 'Yes' by $p_i = P_i(\text{Yes}) / [P_i(\text{Yes}) + P_i(\text{No})]$. We consider the inference correct if (1) the scene falls into the acceptability region and $p_i > 0.5$ or (2) the scene falls out of the acceptability region and $p_i \leq 0.5$.

**Region parsing error.** To mitigate the known bias towards affirmative answers, where $\mathbb{E}[p_i] > 0.5$, we normalize it across all image-prompt pairs, resulting in the normalized probability $\widehat{p}_i := (p_i - \min_j p_j)/(\max_j p_j - \min_j p_j)$. We adopt the root mean square error (RMSE) between the normalized acceptance probability $\widehat{p}$ and reference probability threshold $\lambda^{\text{ref}}$ that represents the actual regions of acceptability, $\varepsilon^{\text{ref}} = \sqrt{\sum_{i=1}^{n} (\widehat{p}_i - \lambda^{\text{ref}})^2/n}$, where $\lambda^{\text{ref}}$ denotes the reference of the assigned probability, analogically to ground-truth labels in machine learning terms.

We introduce two analytically and geometrically motivated proposals defining $\lambda^{\text{ref}}$, $\lambda^{\text{hemi}}$ and $\lambda^{\text{cos}}$, based on hemispheres and cosine of angles, respectively. First, the hemisphere-based reference $\lambda^{\text{hemi}}$ is defined as $\lambda^{\text{hemi}}(\theta) := \mathbb{1}[\theta \in (-90°, 90°)]$. Here, $\theta = 0°$ corresponds to the most prototypical

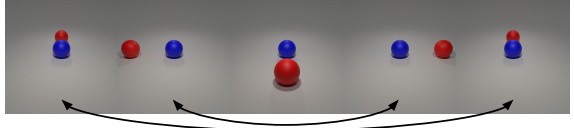 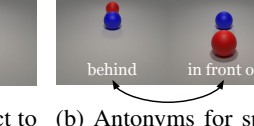 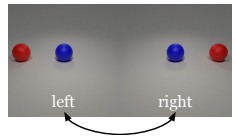

(a) An illustration of the spatial symmetry with respect to the (conventional) front. As the red ball rotates around the blue ball, spatial symmetry consistency ensures that each symmetric pair, with different deviation angles $\theta$ and $-\theta$, has the same probability of being identified as the front.

(b) Antonyms for spatial opposite consistency, e.g., When evaluating if the red ball is to the left of the blue ball, spatial opposite consistency ensures the probability of accepting a sample as left equals the probability of identifying it as not right.

Figure 6: Illustrations for the consistency metrics defined in COMFORT.

spatial relation, and $\theta = 180°$ corresponds to the opposite. Intuitively, $\lambda^{\text{hemi}} = 1$ denotes the test case falls into the acceptable region defined by the textual prompt, and otherwise not. The second reference is derived from the cosine of the deviation angle. Matching the range of the cosine function to that of probability, i.e., $[0, 1]$, we define the cosine-based reference $\lambda^{\cos}(\theta)$ by $\lambda^{\cos}(\theta) := [\cos(\theta) + 1]/2$.

Figure 5 shows an example of the vanilla probability curve $p(\theta)$ from LLaVA-v1.5-7B (Liu et al., 2023b), normalized probability curve $\widehat{p}(\theta)$, and two reference curves $\lambda^{\text{hemi}}(\theta)$ and $\lambda^{\cos}(\theta)$. We report both $\varepsilon^{\text{hemi}}(\theta)$ and $\varepsilon^{\cos}(\theta)$ in experiments. We also note that in human spatial cognition, the regions of acceptability are neither mutually exclusive 90° quadrants nor overlapping 180° hemispheres, as they vary across individuals and depend on the situational context (Franklin et al., 1995).

### 3.5 ROBUSTNESS METRICS

**Standard deviation.** In COMFORT, some images depict variations of the same scene, sharing identical spatial relations between the referent and the object but differing in terms of object colors, sizes, or distractors. When the spatial relation and the query text remain unchanged, an ideal model should have consistent predictions for all variations. To measure model robustness, we report the average standard deviation of the predicted probability $\widehat{p}_i$ across all deviation angles $\sigma := \text{avg}_\theta \sigma(\theta)$.

**Prediction noise.** Since our data is generated through interpolation, ideally, if a model well understands spatial relations, the probability curve with respect to the deviation angle should be low-frequency (i.e., smooth) rather than high-frequency (i.e., noisy). Therefore, we measure the noise by the RMSE, denoted by $\eta$, between the predicted probability and a Butterworth Low Pass Filter (LPF; Butterworth et al., 1930): $\eta := \sqrt{\sum_{i=1}^n [\widehat{p}_i - \text{LPF}(\widehat{p}_i)]^2/n}$. A smaller value of $\eta$ indicates that the probabilities are changing more smoothly, which is more desirable.

### 3.6 CONSISTENCY METRICS

**Spatial symmetric consistency.** A critical aspect of consistent spatial reasoning is geometric symmetry. As our tested target object rotates around the relatum in a circular path that is spatially symmetric, we expect the probabilities of an ideal VLM to consistently reflect geometric symmetry (Figure 6a). For a pair of test cases, indexed by $i$ and $j$, that have the same configurations but opposite deviation angles, i.e., $\theta_i + \theta_j = 0°$, we define the symmetry consistency: $c^{\text{sym}} := \sqrt{2 \sum_{i,j} (\widehat{p}_i - \widehat{p}_j)^2/(n-1)}$.

**Spatial opposite consistency.** Similarly, we expect the probabilities of an ideal VLM to consistently reflect geometric opposition (Figure 6b). For example, the probability that a sample is accepted by the spatial relation "to the left" should be identical to the probability that it is rejected by "to the right." For a pair of opposite spatial relation $r, \text{opp}(r) \in \mathcal{R}$ with the same configurations including the deviation angles $\theta_i$, the opposition consistency is given as: $c^{\text{opp}} := \sqrt{\sum_{i=0}^n (\widehat{p}_i^r + \widehat{p}_i^{\text{opp}(r)} - 1)^2/n}$.

## 4 EMPIRICAL EXPERIMENTS AND MAIN FINDINGS

The COMFORT framework enables us to investigate whether the internal representations of vision-language models encode spatial relations, and if they do, which underlying coordinate systems these representations capture. This can further be broken down into two research questions: (1) When presented with an ambiguous spatial expression, do VLMs follow conventions and exhibit specific preferred FoRs (and the coordinate transformation in relative FoRs) to resolve the ambiguity? (2) How effectively can VLMs adopt different FoRs, when perspectives are explicitly specified to disambiguate spatial expressions paired with visual scenes?

| Model | Back $\varepsilon^{\mathrm{cos}}$ (↓) | | Front $\varepsilon^{\mathrm{cos}}$ (↓) | | Left $\varepsilon^{\mathrm{cos}}$ (↓) | | Right $\varepsilon^{\mathrm{cos}}$ (↓) | | Aggregated | | | Preferred |
|---|---|---|---|---|---|---|---|---|---|---|---|---|
| | Same | **Rev.** | Same | **Rev.** | **Same** | Rev. | **Same** | Rev. | Tran. | Rot. | **Ref.** | |
| InstructBLIP-7B | 45.6 | **39.0** | 31.6 | 52.0 | **37.2** | 48.0 | 47.5 | **37.8** | **40.5** | 44.2 | 43.9 | - |
| InstructBLIP-13B | **40.9** | 45.5 | 46.0 | **37.4** | **43.4** | 44.9 | 45.6 | **41.6** | 44.0 | **42.3** | 43.0 | - |
| mBLIP-BLOOMZ | **51.2** | 53.7 | 51.2 | **47.9** | **52.4** | 53.5 | 54.6 | **46.8** | 52.3 | **50.5** | 52.1 | - |
| GLaMM | 58.3 | **33.3** | 43.9 | **42.9** | **38.3** | 51.8 | **17.3** | 63.7 | 39.5 | 47.9 | **33.0** | **Ref.** |
| LLaVA-1.5-7B | 54.0 | **32.9** | 59.1 | **24.8** | **11.9** | 70.0 | **13.0** | 68.5 | 34.5 | 49.0 | **20.7** | **Ref.** |
| LLaVA-1.5-13B | 61.8 | **19.2** | 56.0 | **27.7** | **31.7** | 61.8 | **24.3** | 64.3 | 43.4 | 43.2 | **25.7** | **Ref.** |
| XComposer2 | 73.2 | **17.9** | 74.5 | **20.7** | **20.1** | 80.9 | **21.3** | 81.1 | 47.3 | 50.1 | **20.0** | **Ref.** |
| MiniCPM-V | 70.9 | **21.9** | 64.3 | **26.9** | **19.7** | 74.1 | **21.1** | 73.3 | 44.0 | 49.1 | **22.4** | **Ref.** |
| GPT-4o | 75.7 | **28.2** | 73.6 | **32.0** | **24.3** | 80.8 | **25.1** | 80.8 | 49.7 | 55.5 | **27.4** | **Ref.** |

Table 2: Preferred coordinate transformation mapping from the egocentric viewer (camera) to the relatum in the relative FoR. The cosine region parsing errors $\varepsilon^{\mathrm{cos}}$ are computed against both the Same and Reversed directions relative to the egocentric viewer's coordinate system. For example, native English speakers typically prefer a Reflected transformation, which maintains the lateral (left/right) axis but reverses the sagittal (front/back) axis relative to the viewer (Figure 1). We determine the preferred transformation based on the aggregated performance, with "-" for no significant preference.

| Model | Back $\varepsilon^{\mathrm{cos}}$ (↓) | | | Front $\varepsilon^{\mathrm{cos}}$ (↓) | | | Left $\varepsilon^{\mathrm{cos}}$ (↓) | | | Right $\varepsilon^{\mathrm{cos}}$ (↓) | | | Aggregated | | | Prefer |
|---|---|---|---|---|---|---|---|---|---|---|---|---|---|---|---|---|
| | **Ego.** | Int. | Add. | **Ego.** | Int. | Add. | **Ego.** | Int. | Add. | **Ego.** | Int. | Add. | **Ego.** | Int. | Add. | |
| InstructBLIP-7B | 41.0 | **38.6** | **38.6** | **40.9** | 46.9 | 46.9 | 45.6 | **32.5** | 51.9 | 39.6 | 51.2 | **31.8** | **41.8** | 42.3 | 42.3 | - |
| InstructBLIP-13B | **32.9** | 34.4 | 34.4 | 52.5 | **48.5** | **48.5** | 47.8 | 56.2 | **27.8** | 40.6 | **27.6** | 56.6 | 43.5 | **41.7** | 41.8 | - |
| mBLIP-BLOOMZ | **52.2** | 53.2 | 53.2 | 45.3 | **44.6** | **44.6** | 47.8 | **47.6** | 48.1 | 45.4 | 48.4 | **42.4** | 47.7 | 48.4 | **47.1** | - |
| GLaMM | **28.0** | 49.1 | 49.1 | **30.0** | 40.2 | 40.2 | **14.0** | 56.8 | 41.5 | **13.7** | 53.0 | 46.6 | **21.4** | 49.8 | 44.4 | **Ego.** |
| LLaVA-1.5-7B | **20.9** | 43.0 | 43.0 | 34.5 | **32.6** | **32.6** | **13.4** | 53.5 | 47.4 | **14.3** | 53.6 | 49.3 | **20.8** | 45.7 | 43.1 | **Ego.** |
| LLaVA-1.5-13B | **31.9** | 38.8 | 38.8 | **24.8** | 57.1 | 57.1 | **11.7** | 51.1 | 51.1 | **27.5** | 57.4 | 48.7 | **24.0** | 51.1 | 48.9 | **Ego.** |
| XComposer2 | **12.7** | 49.3 | 49.3 | **15.2** | 48.3 | 48.3 | **18.8** | 61.2 | 53.7 | **16.5** | 58.4 | 54.5 | **15.8** | 54.3 | 51.4 | **Ego.** |
| MiniCPM-V | **34.2** | 40.7 | 40.7 | **35.5** | 53.4 | 53.4 | **18.0** | 53.9 | 58.4 | **19.0** | 58.1 | 52.7 | **26.7** | 51.5 | 51.3 | **Ego.** |
| GPT-4o | 38.3 | **36.7** | **36.7** | **43.1** | 50.2 | 50.2 | **34.7** | 59.3 | 56.5 | **24.3** | 57.3 | 61.7 | **35.1** | 50.9 | 51.3 | **Ego.** |

Table 3: Preferred frame of reference in VLMs. Models' Cosine Region Parsing Errors $\varepsilon^{\mathrm{cos}}$ are computed against the Intrinsic FoR (relatum origin), Egocentric relative FoR (camera origin), and Addressee-centric relative FoR (addressee origin). English typically prefers an egocentric relative FoR. We determine the preferred FoR based on the aggregated performance, with "-" indicating no significant preference.

In principle, COMFORT can be applied to all VLMs, whether multilingual or monolingual. We note that most existing open-source VLMs are English-based language models; therefore, we begin our experiments on English conventions, where both *relative* and *intrinsic* FoRs are available, but there is a conventional preference for a *relative* FoR combined with a *reflected* coordinate transformation in the relative FoR (see Levinson, 2003, Table 5.4). We further extend our setup to multilingual settings by evaluating models in 109 languages across 170 regions worldwide. To cover a variety of VLMs with different capabilities and training approaches, we evaluate the following models:

- VLMs build from supervised instruction fine-tuning: InstructBLIP-7B/13B- (Dai et al., 2023), LLaVA-v1.5-7B/13B (Liu et al., 2023b), InternLM-XComposer2-7B (Dong et al., 2024);
- VLMs with both supervised fine-tuning and reinforcement learning alignment: MiniCPM-Llama3-V-v2.5-8B (Hu et al., 2024; Yu et al., 2024b);
- Mechanistically grounded VLMs: GLaMM-7B (Rasheed et al., 2024);
- Multilingual VLMs[2]: mBLIP-BLOOMZ-7B (Geigle et al., 2024) and GPT-4o (OpenAI, 2024).

## 4.1 MOST VLMS PREFER REFLECTED COORDINATE TRANSFORMATION CONVENTION

In this section, we address the research question: **do VLMs have a preferred coordinate transformation convention, and if so, what is it?** Experiments are conducted on COMFORT-BALL using the camera perspective prompt (cam) that explicitly specifies an egocentric relative FoR (Table 2). Table 7 in the appendix shows the complete evaluation including $\varepsilon^{\mathrm{hemi}}$ and $\varepsilon^{\mathrm{cos}}$.

We observe that almost all VLMs demonstrate a clear preference over the *reflected* transformation similar to English, except the BLIP series. Still, some models are also affected by the ambiguity of multiple transformation conventions. With the textual prompting specifying a relation, at $\theta = 0$, GPT-4o and LLaVA-1.5-13B show a sharp drop of performance and a significant variance for behind and right, respectively (Figure 7), indicating that some models are sensitive to other transformations.

---

[2]The PaLI series (Chen et al., 2023c;b; 2024b) are closed sourced.

| Model | Egocentric | | Intrinsic | | Addressee | | Aggregated | |
|---|---|---|---|---|---|---|---|---|
| | Acc% (↑) | $\varepsilon^{\cos}_{\times 10^2}(\downarrow)$ | Acc% (↑) | $\varepsilon^{\cos}_{\times 10^2}(\downarrow)$ | Acc% (↑) | $\varepsilon^{\cos}_{\times 10^2}(\downarrow)$ | Acc% (↑) | $\varepsilon^{\cos}_{\times 10^2}(\downarrow)$ |
| InstructBLIP-7B | $47.2_{(+0.0)}$ | $43.5_{(+1.7)}$ | $47.2_{(+0.0)}$ | $42.3_{(+0.0)}$ | $47.2_{(+0.0)}$ | $43.6_{(+1.3)}$ | $47.2_{(+0.0)}$ | $43.1_{(+1.0)}$ |
| InstructBLIP-13B | $47.2_{(+0.0)}$ | $43.8_{(+0.3)}$ | $47.2_{(+0.0)}$ | $43.2_{(+1.5)}$ | $47.2_{(+0.0)}$ | $42.9_{(+1.1)}$ | $47.2_{(+0.0)}$ | $43.3_{(+1.0)}$ |
| mBLIP-BLOOMZ | $51.9_{(-0.9)}$ | $55.4_{(+7.7)}$ | $49.8_{(-3.0)}$ | $54.2_{(+5.8)}$ | $49.6_{(-3.2)}$ | $55.8_{(+8.7)}$ | $50.4_{(-2.4)}$ | $55.1_{(+7.4)}$ |
| GLaMM | $47.2_{(-10.6)}$ | $23.3_{(-0.7)}$ | $47.2_{(+0.8)}$ | $\mathbf{44.2}_{(-6.9)}$ | $47.2_{(-2.8)}$ | $42.8_{(-6.1)}$ | $47.2_{(-4.2)}$ | $36.8_{(-4.6)}$ |
| LLaVA-1.5-7B | $55.2_{(-2.6)}$ | $\mathbf{18.4}_{(-3.0)}$ | $48.3_{(+4.7)}$ | $45.7_{(-4.1)}$ | $48.2_{(-5.0)}$ | $43.4_{(-1.0)}$ | $50.6_{(-1.0)}$ | $\mathbf{35.8}_{(-2.7)}$ |
| LLaVA-1.5-13B | $51.6_{(-15.0)}$ | $23.9_{(+3.1)}$ | $47.3_{(+0.8)}$ | $45.0_{(-0.7)}$ | $47.5_{(-3.8)}$ | $\mathbf{38.9}_{(-4.2)}$ | $48.8_{(-6.0)}$ | $35.9_{(-0.6)}$ |
| XComposer2 | $\mathbf{85.6}_{(-7.0)}$ | $18.8_{(+3.0)}$ | $51.0_{(+0.5)}$ | $51.0_{(-3.3)}$ | $\mathbf{53.2}_{(-0.6)}$ | $49.8_{(-1.6)}$ | $\mathbf{63.3}_{(-2.4)}$ | $39.9_{(-0.6)}$ |
| MiniCPM-V | $72.4_{(-4.8)}$ | $24.6_{(-2.1)}$ | $49.9_{(-2.6)}$ | $47.8_{(-3.7)}$ | $52.9_{(-0.5)}$ | $45.1_{(-6.2)}$ | $58.4_{(-2.6)}$ | $39.2_{(-4.0)}$ |
| GPT-4o | $78.3_{(+4.6)}$ | $28.1_{(-7.0)}$ | $\mathbf{53.4}_{(-1.9)}$ | $44.6_{(-6.3)}$ | $49.1_{(-5.7)}$ | $44.9_{(-6.4)}$ | $60.3_{(-1.0)}$ | $39.2_{(-6.6)}$ |

Table 4: The accuracy and cosine region parsing errors of VLMs when explicitly prompted to follow each frame of reference are provided (cam/rel/add). The values in parentheses indicate the performance change relative to the scenario with no perspective (nop) prompting.

| Model | Obj F1 (↑) | | Acc% (↑) | | $\varepsilon^{\cos}_{\times 10^2}(\downarrow)$ | | $\varepsilon^{\text{hemi}}_{\times 10^2}(\downarrow)$ | | $\sigma_{\times 10^2}(\downarrow)$ | | $\eta_{\times 10^2}(\downarrow)$ | | $c^{\text{sym}}_{\times 10^2}(\downarrow)$ | | $c^{\text{opp}}_{\times 10^2}(\downarrow)$ | |
|---|---|---|---|---|---|---|---|---|---|---|---|---|---|---|---|---|
| | BALL | CAR | BALL | CAR | BALL | CAR | BALL | CAR | BALL | CAR | BALL | CAR | BALL | CAR | BALL | CAR |
| InstructBLIP-7B | 66.7 | 66.7 | 47.2 | 47.2 | 43.9 | 43.5 | 57.8 | 56.4 | 26.7 | 30.5 | 48.4 | 43.4 | 17.2 | 16.9 | 16.6 | 22.6 |
| InstructBLIP-13B | 67.3 | 50.3 | 47.2 | 47.2 | 43.0 | 43.8 | 55.5 | 55.9 | 27.1 | 36.8 | 48.2 | 46.4 | 17.3 | 17.0 | 21.0 | 21.9 |
| mBLIP-BLOOMZ | 99.1 | 33.3 | 47.5 | 51.9 | 52.1 | 55.4 | 62.1 | 65.6 | 43.7 | 48.6 | 54.1 | 60.7 | 29.1 | 30.1 | 33.8 | 42.0 |
| GLaMM | 100.0 | 99.8 | 47.2 | 47.2 | 33.0 | 23.3 | 45.2 | 37.6 | 29.9 | 23.4 | 45.0 | 28.4 | 10.1 | 9.4 | 13.7 | 14.6 |
| LLaVA-1.5-7B | 100.0 | 88.6 | 63.2 | 55.2 | 20.7 | **18.4** | 33.7 | 32.5 | 25.2 | 20.0 | 23.5 | **21.8** | **5.8** | 5.4 | **8.3** | 10.7 |
| LLaVA-1.5-13B | 100.0 | 98.6 | 55.3 | 51.6 | 25.7 | 23.8 | 37.6 | 37.1 | 19.3 | 20.8 | 24.9 | 29.9 | 7.0 | 5.8 | 9.3 | 10.8 |
| XComposer2 | 100.0 | 95.3 | **92.4** | **85.6** | **20.0** | 18.8 | **21.1** | 26.3 | **19.2** | 15.3 | **13.7** | 22.9 | 9.0 | 6.5 | 10.5 | 12.0 |
| MiniCPM-V | 66.8 | 81.5 | 81.0 | 72.4 | 22.4 | 24.6 | 32.8 | 35.8 | 19.2 | **19.2** | 29.8 | 22.7 | 10.1 | 9.2 | 12.4 | 14.9 |
| GPT-4o | 100.0 | 94.5 | 89.2 | 78.3 | 27.4 | 28.1 | 27.5 | 35.0 | 20.9 | 24.0 | 43.1 | 38.8 | 14.1 | 13.3 | 14.2 | 16.7 |
| Random (30 trials) | 50.0 | | 50.9 | | 46.3 | | 58.7 | | 28.3 | | 26.6 | | 42.5 | | 44.2 | |
| Always "Yes" | 50.0 | | 47.2 | | 61.2 | | 68.7 | | 0.0 | | 0.0 | | 0.0 | | 100.0 | |

Table 5: A comprehensive evaluation of VLMs in egocentric relative FoR with reflected transformation, using an explicit camera perspective (cam) prompt, is conducted. The metrics considered include object hallucination (F1-score), accuracy (Acc), region parsing error ($\varepsilon$), prediction noise ($\eta$), standard deviation ($\sigma$), and consistency ($c$).

## 4.2 Most VLMs Prefer Egocentric Relative Frame of Reference

We now attempt to answer the research question: **do VLMs have a preferred frame of reference, and if so, what is it?** We conduct our study on COMFORT-CAR using the no perspective prompt (nop) that deliberately leaves the FoR ambiguous. When calculating the performance with respect to relative FoRs (either egocentric or addressee-centered), we assume a reflected coordinate transformation convention. Table 3 shows the results of preferred FoR in English measured by the region parsing error $\varepsilon^{\cos}$, and Table 8 in the appendix shows the complete eval-

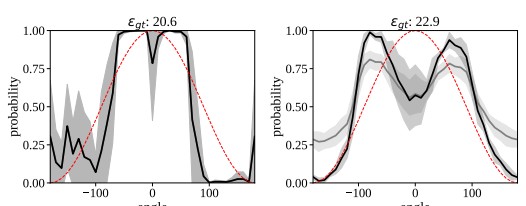

(a) Behind in GPT-4o.  (b) Right in LLaVA-13B.

Figure 7: At $\theta = 0$, some models show sensitivity to multiple conventions.

uation including both $\varepsilon^{\text{hemi}}$ and $\varepsilon^{\cos}$. Almost all VLMs demonstrate a significant preference for the *egocentric relative* FoR similar to English, again, except for the BLIP series. Additionally, the models' performances are inconsistent across different spatial relations—models generally perform better in the lateral directions (left and right) than the sagittal ones (front and behind), even in competitive industry models like GPT-4o. For instance, GLaMM does not show a very strong preference when resolving ambiguities along the sagittal axes, but it demonstrates a significant preference when resolving lateral ambiguity.

## 4.3 VLMs Fail to Adopt Alternative Frames of Reference Flexibly

We now address the research question: **can VLMs adopt different FoRs when perspectives are explicitly specified to disambiguate spatial expressions?** We again use COMFORT-CAR; however, instead of using the no-perspective prompt (nop), we require VLMs to follow one FoR by explicitly specifying the perspective (cam/rel/add) in the textual prompts (Table 1). Table 4 shows the results in accuracy and $\varepsilon^{\cos}$ and the performance compared to when no perspective is specified, and Table 9

| Language | | English | Tamil | Hausa |
|---|---|---|---|---|
| Intrinsic | | 50.9 | 52.0 | 54.0 |
| Ego-Rel | Ref. | **35.8** | **40.4** | **41.0** |
| | Rot. | 57.3 | 55.2 | 56.1 |
| | Tran. | 53.7 | 51.1 | 53.0 |
| Add-Rel | Ref. | 58.8 | 52.2 | 52.8 |
| | Rot. | 51.3 | 52.9 | 55.3 |
| | Tran. | 56.1 | 56.1 | 56.1 |
| GPT-4o Prefer | | Ego-Ref. | Ego-Ref. | Ego-Ref. |
| Human Prefer | | Ego-Ref. | Ego-Rot. | Ego-Trans. |

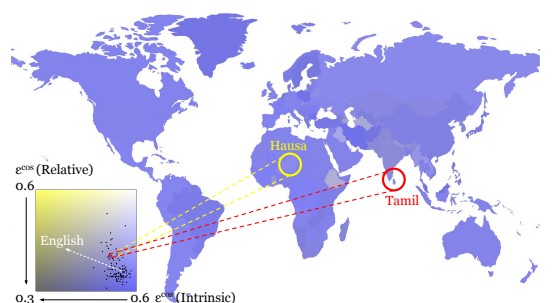

Figure 8: GPT-4o's preferences for intrinsic FoR over the relative FoR across regions. The plot is based on the top three spoken languages in each region, as reported by The World Factbook (Central Intelligence Agency, 2009), and averages the cosine parsing error ($\varepsilon^{\text{cos}}$, $\downarrow$), weighted by the speaking population. We present a quantitative comparison of English, Tamil, and Hausa, with the best-performing FoR marked in bold and the convention preferred by human speakers underlined.

in the appendix gives the complete evaluation. We find that all models, including the strong ones like GPT-4 and InternLM-XComposer2, show close-to-chance performance (50% accuracy) when being prompted to use the intrinsic or addressee-centered relative FoRs. Compared to the same probing setup without a perspective specified (nop), we find generally marginal improvements in region parsing error ($\varepsilon$), whereas the accuracy decreases. Overall, the results indicate that while VLMs can comprehend scenes using egocentric relative FoR, they struggle to adapt flexibly to alternative FoRs.

## 4.4 SPATIAL REPRESENTATIONS IN VLMS ARE NOT ROBUST AND CONSISTENT

In this section, we further ask: **are spatial representations in VLMs robust and consistent?** The considered metrics include accuracy (Acc), region parsing error ($\varepsilon$), prediction noise ($\eta$), standard deviation ($\sigma$), and consistency ($c$) as defined in Section 3. One commonly considered possibility that VLMs underperform is that they suffer from *object hallucination*, where they misperceive objects in the scenes (Li et al., 2023; Chen et al., 2024c). Following the object probing setups, we prompt the VLMs to inquire about the presence of an existing object and a non-existing object in the scene, and compute the F1-score (Table 5). We find that the BLIP models suffer from severe object hallucinations, which contribute to their underperformance in the previous evaluation. Many VLMs, despite showing decent performance metrics in terms of spatial understanding and reasoning accuracy, demonstrate a lack of robustness and consistency. For example, the spatial opposite consistency ($c^{\text{opp}}$) of GPT-4 is not significantly better than 30 random trials. In contrast, VLMs that have undergone supervised fine-tuning on spatial relation tasks have a more robust and consistent spatial representation. For instance, InternLM-XComposer2 and MiniCPM-V (on the COMFORT-BALL task, with no object hallucinations) show improved performance. On the other hand, although GLaMM is mechanistically grounded to objects and exhibits minimal object hallucination, its spatial understanding capability is poor. This suggests that improving visual entity grounding helps in recognizing individual objects but does not automatically translate to better spatial understanding between multiple objects.

## 4.5 A CROSS-LINGUAL AND CROSS-CULTURAL EVALUATION OF FRAME OF REFERENCE

All previous experiments are centered around English; however, individuals from diverse multilingual and cultural backgrounds may adopt different preferences and conventions to select their FoR in resolving ambiguities (Majid et al., 2004; O'Meara & Báez, 2011; Bohnemeyer et al., 2014; Ogelo & Bylund, 2024). Our next research question naturally arises: **Do multilingual VLMs faithfully follow the preferences and conventions (associated with different languages) to select the FoR?** To extend the study of preferred FoR from English to a multilingual setting, we evaluate 109 languages worldwide to investigate whether each language shows a preferred FoR. We translate the English prompts into the target languages using the Google Cloud Translate API. Given that the open-source language models either lack strong multilingual capabilities or underperform in previous evaluations, we study this problem on the GPT-4o model (OpenAI, 2024). We follow the setup similar to Section 4.2, but only evaluate the images corresponding to the four canonical directions using the nop prompt. For each language, we compute $\varepsilon^{\text{cos}}$ for each FoR and coordinate transformation. Figure 8 presents a visualization of the world map, displaying the preference of each region for using the (object-centered) intrinsic FoR over the relative FoR, where the latter corresponds to a low $\varepsilon^{\text{cos}}$ value. Table 10 in the Appendix summarizes the results across all tested languages.

Nearly all tested languages demonstrate a preference towards the relative FoR, except several underrepresented languages, such as Konkani, Kurdish, and Amharic, which exhibit near-random performance without a significant preference. In Figure 8, we present a classic comparison between English, Tamil, and Hausa similar to that of Levinson (2003), with the best-performing FoR marked in bold, and the preferred convention by humans underlined. Although human speakers of these languages have different preferred coordinate transformation conventions, the English convention of reflected projection is observed for both Tamil and Hausa. Although, for example, Hausa permits an English-like interpretation of front-back relations, this interpretation is generally less favored and may confuse Hausa speakers (Hill, 1982). This raises concerns that English may dominate the FoR preference conventions of other languages in multilingual VLMs.

## 5 DISCUSSIONS

**Do vision-language models represent space and how?** It is insufficient to answer this question by simply querying the model with text-image pairs and comparing the output with a fixed ground truth. We must, at least, query the models with awareness of the ambiguity in FoRs, which is essential in determining how the scenes in the physical world are mapped to spatial expressions (Levinson, 2003). Our experiments confirm that many VLMs are equipped with reasonable spatial representations through vision-language training alone; in particular, most VLMs clearly prefer the egocentric relative FoR with reflected projection, aligning with English conventions. However, our results also show these representations lack robustness and consistency in a continuous space. Similar experimental setups can yield widely varying performance across different spatial relations—for example, GPT-4 shows minimal preferences for the egocentric relative FoR along the sagittal axis but a significant preference along the lateral one (Table 3). As a result, VLMs demonstrate unsatisfactory consistency in their spatial performance (Table 5). Future work is necessary to improve the consistency and robustness of spatial representations in these models.

**Perspective taking as a prerequisite of human-like spatial reasoning.** Most languages support multiple FoRs.[3] The ability to understand and reason about space from a non-egocentric perspective is an important foundation of the Theory of Mind, a basic building block of our situated communication skill that allows us to infer others' mental states (Ma et al., 2023b). One of our key findings is that VLMs still struggle to adopt alternative FoRs flexibly, even when provided with explicit perspective-taking instructions (Section 4.3). We hypothesize that this phenomenon may come from a reporting bias in the image-text datasets available on the internet—it is natural to take the reflected relative FoR to view images presented on a screen, but this does not always apply in real-world applications. To address this issue, we suggest future work extend the current 2D VLMs to the 3D domain, by considering camera poses and multiview data (Yang et al., 2024; Hao et al., 2024) for training.

**Cross-cultural conventions in cross-lingual spatial understanding.** The conventions for resolving spatial ambiguities are not uniform, as individuals from diverse linguistic and cultural backgrounds select their FoR differently. Cultural conventions can even be transmitted as individuals are exposed to other languages. Bohnemeyer et al. (2014) found that among indigenous language speakers with various preferences in FoRs, those more proficient in Spanish tend to use the reflected relative FoR (Spanish convention) more in their native language. This phenomenon has led to their Linguistic Transmission Hypothesis: "Using any language or linguistic variety – independently of its structures – may facilitate the acquisition of cultural practices of nonlinguistic cognition shared among the speakers of the language." Analogously, our experiment raises important concerns that English may dominate the FoR preference conventions of other languages in multilingual VLMs. This is not surprising, as current training recipes for multilingual multimodal language models heavily rely on machine-translated captions (Chen et al., 2023c; Geigle et al., 2024); however, this practice can be problematic: for instance, Hausa prefers an interpretation where the "front" aligns with the English concept of "back," (Hill, 1982), where this approach may lead to English conventions overshadowing those of other languages. At a high level, this issue is not limited to spatial reasoning: as an example, Shi et al. (2023) have demonstrated that English is always the best chain-of-thought language for math reasoning with multilingual LLMs, no matter what language is used for the problem description. To enable similar linguistic transmission in AI models, exposure to naturally generated multilingual image-text data is crucial (Romero et al., 2024).

---

[3]Some languages, in very rare cases, have only one available spatial frame of reference. For example, Guugu Yimithirr exclusively uses the absolute FoR (Levinson, 2003).

ACKNOWLEDGMENTS

This work was supported in part by NSF IIS-1949634, NSERC RGPIN-2024-04395, ONR N00014-23-1-2417, and the Microsoft Accelerate Foundation Models Research (AFMR) grant program. Ziqiao Ma is supported in part by the Weinberg Cognitive Science Fellowship. Any opinions, findings, conclusions, or recommendations expressed in this material are those of the authors and do not necessarily reflect the views of these funding agencies. The authors would like to thank Yinpei Dai, Run Peng, Jung-Chun Liu, and Xuejun Zhang for proofreading and feedback.

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

# A DATASET AND METRIC DETAILS

## A.1 DATASET CONFIGURATIONS

The entire data generation pipeline produces 720 English test cases in COMFORT-BALL, and 57.6k English test cases in COMFORT-CAR. For COMFORT-BALL: 1 object combination × 5 variants × 4 relations × 36 angles = 720 test cases. For COMFORT-CAR: 20 object combinations × 5 variants × 4 relations × 36 angles × 4 prompts = 57,600 test cases. The table below lists all possible variants and configurations for the dataset, and we describe our dataset configuration in detail as follows.

| Test Case Setup | Possible Variants |
|---|---|
| Scene $\mathcal{S}$ | COMFORT-BALL: **Relatum**: *red ball*; **Referent**: *blue ball*; 36 samples uniformly collected along a rotational path. |
| | COMFORT-CAR: **Relatum**: *basketball*; **Referent**: *horse*, *car*, *bench*, *laptop*, *rubber duck*, *chair*, *dog*, *sofa*, *bed*, *bicycle*; **Addressee**: woman; 36 samples uniformly collected along a rotational path. |
| Spatial Relation $\mathcal{R}$ | *to the left of*, *to the right of*, *in front of*, *behind* |
| Frame of Reference $\mathcal{F}$ | *egocentric relative*, *addressee-centered relative*, *object-centered intrinsic* |
| Language $\mathcal{L}$ | See Table 10. |

Table 6: A test case in COMFORT is defined as a 4-tuple in $\mathcal{S} \times \mathcal{R} \times (\mathcal{F} \cup \{\emptyset\}) \times \mathcal{L}$. This table enumerates all possible variants and configurations of the dataset.

## A.2 LIST OF EVALUATED LANGUAGES

We started with 132 candidate languages supported by Google Translate API.[4] We removed 23 languages from our multilingual evaluation due to their failure to adhere to instructions for generating "yes" and "no" predictions, or because they did not pass the back-translation test for quality control: Aymara, Bambara, Croatian, Dhivehi, Dogri, Ewe, Guarani, Hmong, Kyrgyz, Luganda, Malayalam, Meiteilon (Manipuri), Mizo, Odia (Oriya), Punjabi, Quechua, Samoan, Tatar, Telugu, Tigrinya, Uyghur, Xhosa, Yoruba.

## A.3 VISUALIZATIONS OF REGION PARSING ERROR

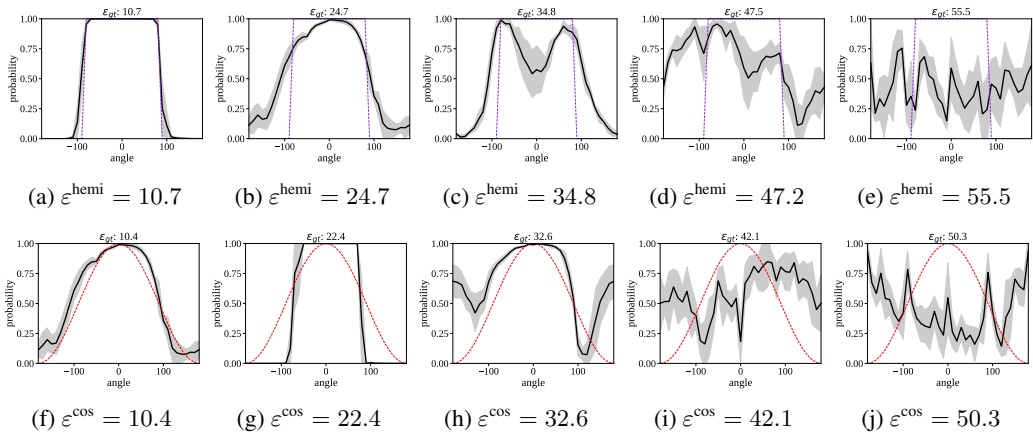

(a) $\varepsilon^{\mathrm{hemi}} = 10.7$    (b) $\varepsilon^{\mathrm{hemi}} = 24.7$    (c) $\varepsilon^{\mathrm{hemi}} = 34.8$    (d) $\varepsilon^{\mathrm{hemi}} = 47.2$    (e) $\varepsilon^{\mathrm{hemi}} = 55.5$

(f) $\varepsilon^{\cos} = 10.4$    (g) $\varepsilon^{\cos} = 22.4$    (h) $\varepsilon^{\cos} = 32.6$    (i) $\varepsilon^{\cos} = 42.1$    (j) $\varepsilon^{\cos} = 50.3$

Figure 9: $\varepsilon$ visualization: (a-e) correspond to $\varepsilon^{\mathrm{hemi}}$, and (f-j) correspond to $\varepsilon^{\cos}$.

# B LIMITATIONS AND INTENDED USE

## B.1 LIMITATIONS

**The acceptance regions.** Using cosine and hemisphere as acceptance regions is analytical but might not capture some human cognitive biases. In reality, regional angles might not be uniformly

---

[4]https://cloud.google.com/translate

distributed per relation, nor are they exactly 90 degrees. These angles vary across individuals and cultures (Franklin et al., 1995).

**Spatial relations.** This work primarily focuses on the most basic types of spatial relations (front-back and left-right). However, many other relations exist, such as *away from* and *near* (Logan & Sadler, 1996; Liu et al., 2023a). Additionally, not all languages possess terms for "left," "right," "front," and "back." Some languages, like Guugu Yimithirr, use only absolute frames of reference instead (Levinson, 2003).

**Camera angle and occlusion.** Currently, there is no occlusion, and the camera angle is high. Languages may differ in the importance placed on these factors, such as the preference to use "behind" in cases of occlusion (Levinson, 2003).

**Pragmatic aspect of spatial cognition.** Many conversational and pragmatic aspects of spatial cognition are simplified in this work, such as F-Formation (Kendon, 2010) and human-robot interaction (Liu et al., 2010). For example, in human-robot interaction settings, users prefer an addressee-centered frame of reference to facilitate the robot's comprehension of spatial referents (Moratz & Tenbrink, 2006).

**Multilingual prompts.** In this work, we used machine-translated text to construct the multilingual portion of the dataset. Although we verified data quality through back translation, incorporating human annotations in the future would be a valuable future step.

## B.2 INTENDED USE

We intentionally use the term "framework" to emphasize that this dataset and its associated evaluation metrics are designed to assess cognitive similarity and alignment with human spatial cognition. While each studied language demonstrates a preference, we do not position this as a leaderboard-driven benchmark. However, the perspective-taking capability of VLMs as studied in § 4.3 can function as a benchmark, as the input and ground truth are unambiguous. We encourage future work in VLM training to address this specific challenge.

## C CASE STUDIES

We added two case studies to augment the results in COMFORT:

### C.1 ALTERNATIVE BACKGROUND

We first create a case study with a brown background color brown from COMFORT-BALL. We keep the objects visible and generate 36 images for each spatial relation. The images are shown below:

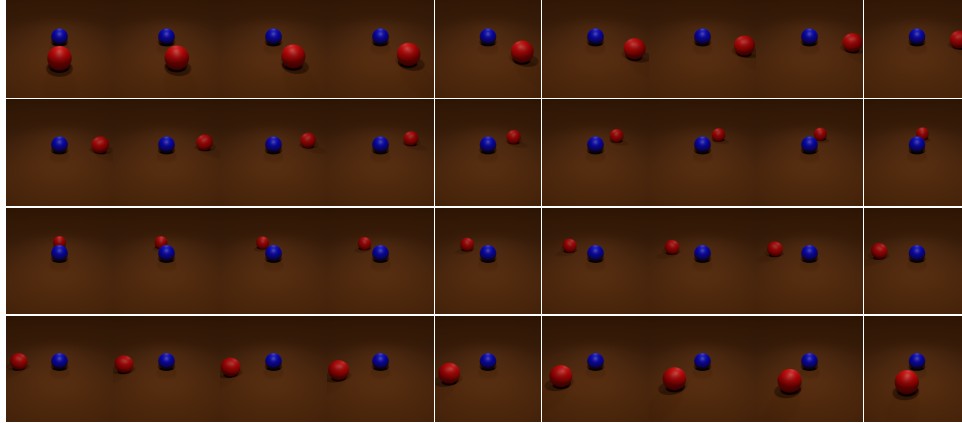

### C.2 REAL-WORLD IMAGES

We added a pair of case studies to see if results in synthetic images generalize to real images. We first create a real-world version of the COMFORT-BALL dataset with no variations. We fixed the camera pose

and manually rotated the target object around the reference object using two equally sized red and blue balls, with all other setups identical to the Blender simulation. We took 36 images in total along the rotational path for each spatial relation. We also create a real-world version of the COMFORT-CAR dataset without the addressee. We used a laptop and the red ball to collect a set of 36 images for each spatial relation. The images are shown below:

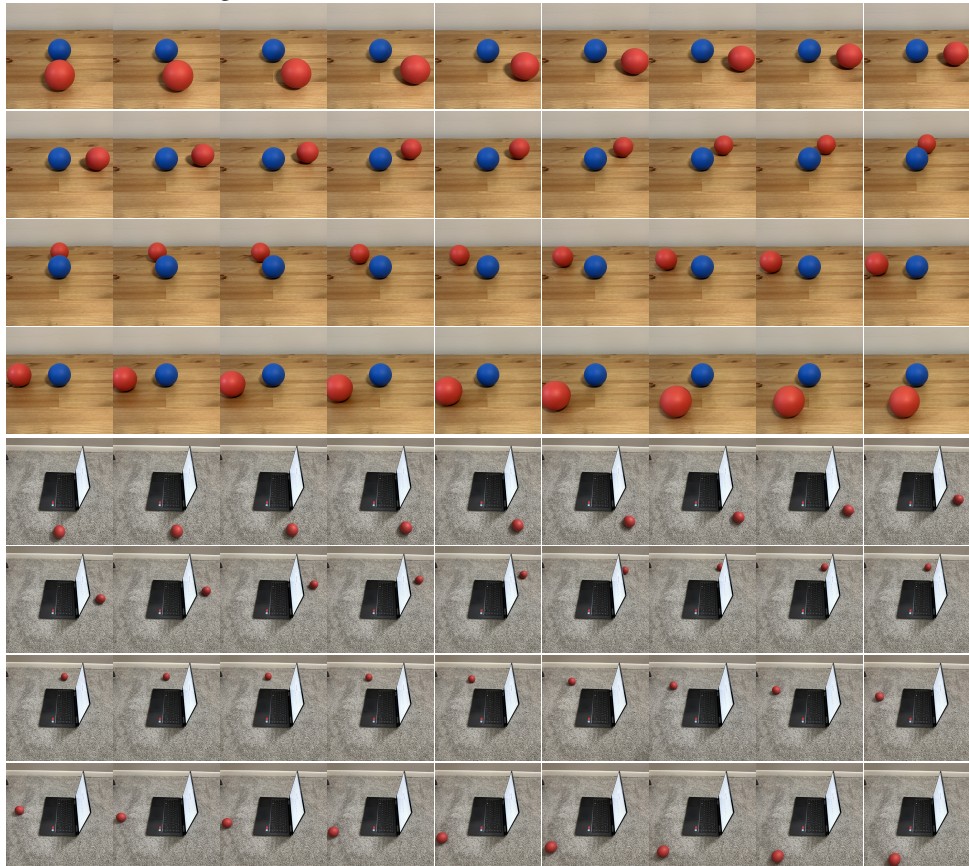

## C.3    RESULTS

For the case study, we evaluated three VLMs: LLaVA-v1.5-7B/13B Liu et al. (2023b) and GPT-4o OpenAI (2024). Prediction plots under the camera perspective prompt (cam) are shown in Figure 10. Prediction plots under the camera perspective prompt (cam) are shown in Figure 11. Prediction plots under the camera perspective prompt (cam) are shown in Figure 12. Our findings remain consistent even when applied to these case studies and the general trend holds. However, we did observe that models performed with more noise compared to the synthetic data. We hypothesize that this noise arises from small inconsistencies between images in the real dataset, such as slight tilts in the positioning of the camera and the target object.

## D    ADDENDUM TO RESULTS

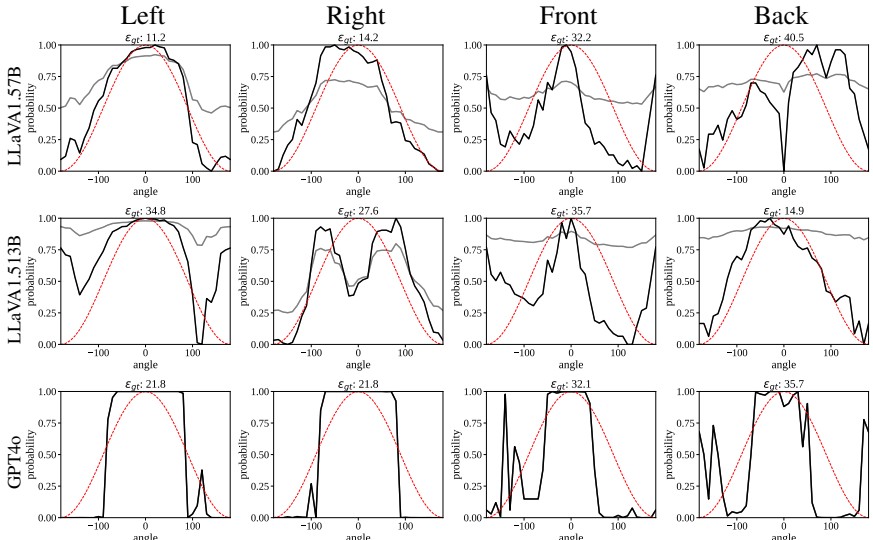

Figure 10: All prediction plots for each model on COMFORT-BALL with brown background using the camera perspective prompt (cam). The raw probability $p(\theta)$ in gray, normalized probability $\widehat{p}(\theta)$ in black, and the reference probability $p_{\cos}(\theta)$ of cam in red.

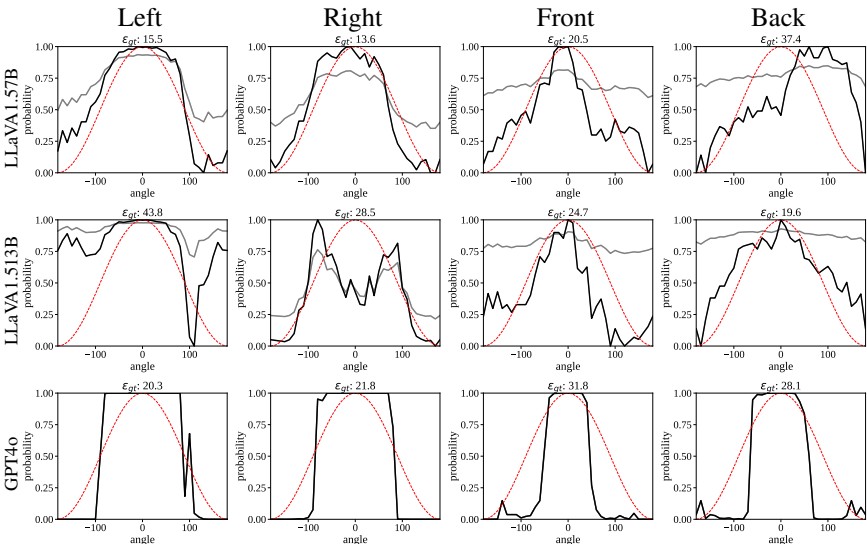

Figure 11: All prediction plots for each model on real image version of COMFORT-BALL using the camera perspective prompt (cam). The raw probability $p(\theta)$ in gray, normalized probability $\widehat{p}(\theta)$ in black, and the reference probability $p_{\cos}(\theta)$ of cam in red.

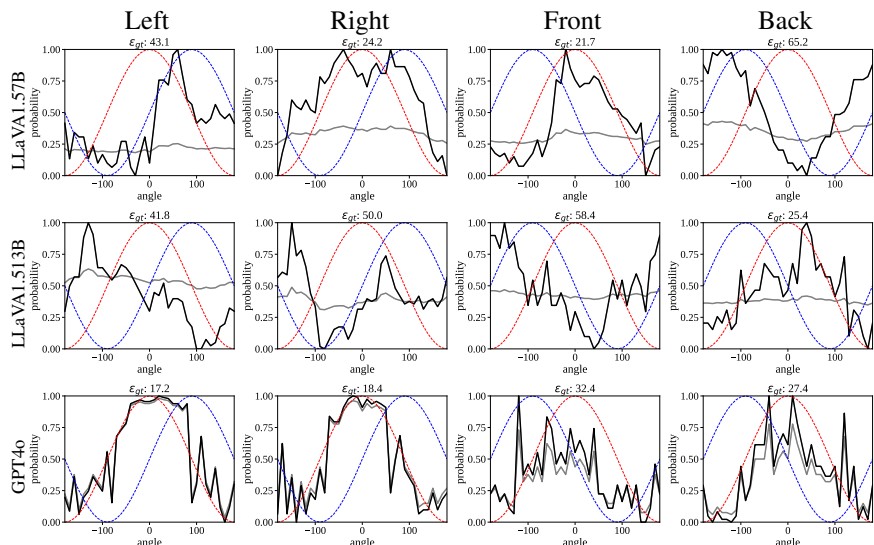

Figure 12: All prediction plots for each model on the real image version of COMFORT-CAR using the camera perspective prompt (cam). The raw probability $p(\theta)$ in gray, normalized probability $\widehat{p}(\theta)$ in black, and the reference probability $p_{\cos}(\theta)$ of cam in red, rel in blue.

| | Back | | | | | | Front | | | | | |
|---|---|---|---|---|---|---|---|---|---|---|---|---|
| | Same | | | Reversed | | | Same | | | Reversed | | |
| | Acc% | $\varepsilon^{\text{hemi}}_{\times10^2}$ | $\varepsilon^{\cos}_{\times10^2}$ | Acc% | $\varepsilon^{\text{hemi}}_{\times10^2}$ | $\varepsilon^{\cos}_{\times10^2}$ | Acc% | $\varepsilon^{\text{hemi}}_{\times10^2}$ | $\varepsilon^{\cos}_{\times10^2}$ | Acc% | $\varepsilon^{\text{hemi}}_{\times10^2}$ | $\varepsilon^{\cos}_{\times10^2}$ |
| InstructBLIP-7B | 47.2 | 58.4 | 45.6 | 47.2 | 53.8 | 39.0 | 47.2 | 47.5 | 31.6 | 47.2 | 64.6 | 52.0 |
| InstructBLIP-13B | 47.2 | 55.9 | 40.9 | 47.2 | 56.6 | 45.5 | 47.2 | 60.0 | 46.0 | 47.2 | 53.0 | 37.4 |
| mBLIP | 56.1 | 60.2 | 51.2 | 47.2 | 64.8 | 53.7 | 51.1 | 61.4 | 51.2 | 47.8 | 58.0 | 47.9 |
| GLaMM | 47.2 | 71.1 | 58.3 | 47.2 | 46.3 | 33.3 | 47.2 | 55.4 | 43.9 | 47.2 | 55.9 | 42.9 |
| LLaVA-1.5-7B | 47.2 | 66.7 | 54.0 | 47.2 | 47.0 | 32.9 | 47.2 | 71.0 | 59.1 | 47.2 | 36.4 | 24.8 |
| LLaVA-1.5-13B | 47.2 | 73.8 | 61.8 | 47.2 | 36.3 | 19.2 | 42.8 | 67.3 | 56.0 | 51.7 | 39.1 | 27.7 |
| XComposer2 | 13.3 | 84.5 | 73.2 | 90.0 | 26.3 | 17.9 | 15.0 | 85.8 | 74.5 | 85.0 | 31.6 | 20.7 |
| MiniCPM-V | 13.9 | 84.1 | 70.9 | 80.6 | 35.6 | 21.9 | 26.1 | 77.0 | 64.3 | 75.0 | 35.3 | 26.9 |
| GPT-4o | 16.7 | 87.3 | 75.7 | 87.8 | 30.3 | 28.2 | 25.6 | 82.4 | 73.6 | 80.0 | 40.2 | 32.0 |
| | Left | | | | | | Right | | | | | |
| | Same | | | Reversed | | | Same | | | Reversed | | |
| | Acc% | $\varepsilon^{\text{hemi}}_{\times10^2}$ | $\varepsilon^{\cos}_{\times10^2}$ | Acc% | $\varepsilon^{\text{hemi}}_{\times10^2}$ | $\varepsilon^{\cos}_{\times10^2}$ | Acc% | $\varepsilon^{\text{hemi}}_{\times10^2}$ | $\varepsilon^{\cos}_{\times10^2}$ | Acc% | $\varepsilon^{\text{hemi}}_{\times10^2}$ | $\varepsilon^{\cos}_{\times10^2}$ |
| InstructBLIP-7B | 47.2 | 51.5 | 37.2 | 47.2 | 61.6 | 48.0 | 47.2 | 61.4 | 47.5 | 47.2 | 52.0 | 37.8 |
| InstructBLIP-13B | 47.2 | 54.2 | 43.4 | 47.2 | 57.0 | 44.9 | 47.2 | 58.1 | 45.6 | 47.2 | 52.5 | 41.6 |
| mBLIP | 47.2 | 59.8 | 52.4 | 47.2 | 64.2 | 53.5 | 47.8 | 65.7 | 54.6 | 47.8 | 56.4 | 46.8 |
| GLaMM | 47.2 | 48.9 | 38.3 | 47.2 | 65.5 | 51.8 | 79.4 | 29.8 | 17.3 | 15.0 | 76.2 | 63.7 |
| LLaVA-1.5-7B | 47.2 | 25.3 | 11.9 | 47.2 | 83.4 | 70.0 | 47.2 | 26.0 | 13.0 | 47.2 | 80.9 | 68.5 |
| LLaVA-1.5-13B | 62.8 | 39.1 | 31.7 | 31.7 | 76.8 | 61.8 | 91.1 | 35.8 | 24.3 | 8.9 | 79.3 | 64.3 |
| XComposer2 | 97.8 | 11.3 | 20.1 | 3.3 | 95.6 | 80.9 | 96.7 | 15.2 | 21.3 | 3.3 | 95.8 | 81.1 |
| MiniCPM-V | 86.1 | 27.7 | 19.7 | 9.4 | 88.1 | 74.1 | 82.2 | 32.7 | 21.1 | 12.2 | 87.0 | 73.3 |
| GPT-4o | 94.4 | 20.4 | 24.3 | 11.1 | 92.6 | 80.8 | 94.4 | 19.0 | 25.1 | 11.1 | 92.8 | 80.8 |
| | Aggregated | | | | | | | | | | | |
| | Translated | | | Rotated | | | Reflected | | | Preferred Transform | | |
| | Acc% | $\varepsilon^{\text{hemi}}_{\times10^2}$ | $\varepsilon^{\cos}_{\times10^2}$ | Acc% | $\varepsilon^{\text{hemi}}_{\times10^2}$ | $\varepsilon^{\cos}_{\times10^2}$ | Acc% | $\varepsilon^{\text{hemi}}_{\times10^2}$ | $\varepsilon^{\cos}_{\times10^2}$ | | | |
| InstructBLIP-7B | 47.2 | 54.7 | 40.5 | 47.2 | 58.0 | 44.2 | 47.2 | 57.8 | 43.9 | Not Significant | | |
| InstructBLIP-13B | 47.2 | 57.1 | 44.0 | 47.2 | 54.8 | 42.3 | 47.2 | 55.5 | 43.0 | Not Significant | | |
| mBLIP | 50.6 | 61.8 | 52.3 | 47.5 | 60.9 | 50.5 | 47.5 | 62.1 | 52.1 | Not Significant | | |
| GLaMM | 55.3 | 51.3 | 39.5 | 39.2 | 61.0 | 47.9 | 55.3 | 45.2 | 33.0 | Reflected | | |
| LLaVA-1.5-7B | 47.2 | 47.3 | 34.5 | 47.2 | 61.9 | 49.0 | 47.2 | 33.7 | 20.7 | Reflected | | |
| LLaVA-1.5-13B | 61.0 | 54.0 | 43.4 | 34.9 | 57.9 | 43.2 | 63.2 | 37.6 | 25.7 | Reflected | | |
| XComposer2 | 55.7 | 49.2 | 47.3 | 45.4 | 62.3 | 50.1 | 92.4 | 21.1 | 20.0 | Reflected | | |
| MiniCPM-V | 52.1 | 55.4 | 44.0 | 44.3 | 61.5 | 49.1 | 81.0 | 32.8 | 22.4 | Reflected | | |
| GPT-4o | 57.8 | 52.3 | 49.7 | 47.5 | 64.0 | 55.5 | 89.2 | 27.5 | 27.4 | Reflected | | |

Table 7: The full results for testing the preferred coordinate transformation mapping from the viewer to the relatum in the relative frame of reference.

| | Back Egocentric | | | Back Intrinsic | | | Back Addressee | | | Front Egocentric | | | Front Intrinsic | | | Front Addressee | | |
|---|---|---|---|---|---|---|---|---|---|---|---|---|---|---|---|---|---|---|
| | Acc% | $\varepsilon^{hemi}_{\times10^2}$ | $\varepsilon^{cos}_{\times10^2}$ | Acc% | $\varepsilon^{hemi}_{\times10^2}$ | $\varepsilon^{cos}_{\times10^2}$ | Acc% | $\varepsilon^{hemi}_{\times10^2}$ | $\varepsilon^{cos}_{\times10^2}$ | Acc% | $\varepsilon^{hemi}_{\times10^2}$ | $\varepsilon^{cos}_{\times10^2}$ | Acc% | $\varepsilon^{hemi}_{\times10^2}$ | $\varepsilon^{cos}_{\times10^2}$ | Acc% | $\varepsilon^{hemi}_{\times10^2}$ | $\varepsilon^{cos}_{\times10^2}$ |
| InstructBLIP-7B | 47.2 | 51.4 | 41.0 | 47.2 | 53.0 | 38.6 | 47.2 | 53.0 | 38.6 | 47.2 | 54.2 | 40.9 | 47.2 | 60.7 | 46.9 | 47.2 | 60.7 | 46.9 |
| InstructBLIP-13B | 47.2 | 43.5 | 32.9 | 47.2 | 48.9 | 34.4 | 47.2 | 48.9 | 34.4 | 47.2 | 66.5 | 52.5 | 47.2 | 61.1 | 48.5 | 47.2 | 61.1 | 48.5 |
| mBLIP-BLOOMZ | 52.8 | 62.1 | 52.2 | 52.8 | 63.9 | 53.2 | 52.8 | 63.9 | 53.2 | 52.8 | 56.4 | 45.3 | 52.8 | 55.5 | 44.6 | 52.8 | 55.5 | 44.6 |
| GLaMM | 47.2 | 45.6 | 31.9 | 47.2 | 51.0 | 38.8 | 47.2 | 51.0 | 38.8 | 47.2 | 37.9 | 24.8 | 47.2 | 69.6 | 57.1 | 47.2 | 69.6 | 57.1 |
| LLaVA-1.5-7B | 49.2 | 41.6 | 28.0 | 47.5 | 60.3 | 49.1 | 47.5 | 60.3 | 49.1 | 48.6 | 43.2 | 30.0 | 48.6 | 52.9 | 40.2 | 48.6 | 52.9 | 40.2 |
| LLaVA-1.5-13B | 50.8 | 36.8 | 20.9 | 48.6 | 54.7 | 43.0 | 48.6 | 54.7 | 43.0 | 47.2 | 46.5 | 34.5 | 47.2 | 47.3 | 32.6 | 47.2 | 47.3 | 32.6 |
| XComposer2 | 91.4 | 25.0 | 12.7 | 53.6 | 59.9 | 49.3 | 53.6 | 59.9 | 49.3 | 87.8 | 26.6 | 15.2 | 55.0 | 59.3 | 48.3 | 55.0 | 59.3 | 48.3 |
| MiniCPM-V | 66.4 | 46.5 | 34.2 | 60.8 | 51.3 | 40.7 | 60.8 | 51.3 | 40.7 | 57.5 | 45.0 | 35.5 | 50.8 | 64.6 | 53.4 | 50.8 | 64.6 | 53.4 |
| GPT-4o | 64.2 | 49.1 | 38.3 | 66.4 | 45.4 | 36.7 | 66.4 | 45.4 | 36.7 | 58.1 | 54.8 | 43.1 | 53.6 | 61.0 | 50.2 | 53.6 | 61.0 | 50.2 |

| | Left Egocentric | | | Left Intrinsic | | | Left Addressee | | | Right Egocentric | | | Right Intrinsic | | | Right Addressee | | |
|---|---|---|---|---|---|---|---|---|---|---|---|---|---|---|---|---|---|---|
| | Acc% | $\varepsilon^{hemi}_{\times10^2}$ | $\varepsilon^{cos}_{\times10^2}$ | Acc% | $\varepsilon^{hemi}_{\times10^2}$ | $\varepsilon^{cos}_{\times10^2}$ | Acc% | $\varepsilon^{hemi}_{\times10^2}$ | $\varepsilon^{cos}_{\times10^2}$ | Acc% | $\varepsilon^{hemi}_{\times10^2}$ | $\varepsilon^{cos}_{\times10^2}$ | Acc% | $\varepsilon^{hemi}_{\times10^2}$ | $\varepsilon^{cos}_{\times10^2}$ | Acc% | $\varepsilon^{hemi}_{\times10^2}$ | $\varepsilon^{cos}_{\times10^2}$ |
| InstructBLIP-7B | 47.2 | 59.0 | 45.6 | 47.2 | 45.3 | 32.5 | 47.2 | 62.0 | 51.9 | 47.2 | 53.1 | 39.6 | 47.2 | 61.7 | 51.2 | 47.2 | 45.3 | 31.8 |
| InstructBLIP-13B | 47.2 | 59.7 | 47.8 | 47.2 | 70.2 | 56.2 | 47.2 | 39.6 | 27.8 | 47.2 | 53.6 | 40.6 | 47.2 | 39.5 | 27.6 | 47.2 | 70.8 | 56.6 |
| mBLIP-BLOOMZ | 52.8 | 58.2 | 47.8 | 52.8 | 59.7 | 47.6 | 52.8 | 58.4 | 48.1 | 52.8 | 57.7 | 45.4 | 52.8 | 60.6 | 48.4 | 52.8 | 53.8 | 42.4 |
| GLaMM | 75.8 | 22.3 | 11.7 | 46.4 | 62.0 | 51.1 | 52.5 | 62.3 | 51.1 | 60.8 | 41.8 | 27.5 | 44.7 | 68.5 | 57.4 | 53.1 | 58.7 | 48.7 |
| LLaVA-1.5-7B | 76.7 | 25.6 | 14.0 | 33.9 | 68.2 | 56.8 | 64.4 | 52.7 | 41.5 | 56.4 | 28.5 | 13.7 | 44.2 | 64.6 | 53.0 | 52.5 | 57.3 | 46.6 |
| LLaVA-1.5-13B | 81.7 | 23.7 | 13.4 | 42.2 | 65.0 | 53.5 | 57.2 | 58.5 | 47.4 | 86.7 | 26.8 | 14.3 | 47.8 | 64.0 | 53.6 | 52.2 | 59.9 | 49.3 |
| XComposer2 | 95.0 | 18.8 | 18.8 | 45.6 | 70.5 | 61.2 | 54.4 | 64.0 | 53.7 | 96.1 | 17.1 | 16.5 | 47.8 | 68.1 | 58.4 | 52.2 | 64.6 | 54.5 |
| MiniCPM-V | 93.3 | 20.4 | 18.0 | 52.2 | 64.3 | 53.9 | 47.8 | 68.0 | 58.4 | 91.7 | 22.6 | 19.0 | 46.1 | 68.3 | 58.1 | 53.9 | 62.5 | 52.7 |
| GPT-4o | 78.6 | 42.1 | 34.7 | 48.1 | 69.4 | 59.3 | 51.9 | 65.8 | 56.5 | 93.9 | 21.8 | 24.3 | 52.8 | 67.0 | 57.3 | 47.2 | 71.0 | 61.7 |

| | Aggregated Egocentric | | | Aggregated Intrinsic | | | Aggregated Addressee | | | Preferred FoR |
|---|---|---|---|---|---|---|---|---|---|---|
| | Acc% | $\varepsilon^{hemi}_{\times10^2}$ | $\varepsilon^{cos}_{\times10^2}$ | Acc% | $\varepsilon^{hemi}_{\times10^2}$ | $\varepsilon^{cos}_{\times10^2}$ | Acc% | $\varepsilon^{hemi}_{\times10^2}$ | $\varepsilon^{cos}_{\times10^2}$ | |
| InstructBLIP-7B | 47.2 | 54.4 | 41.8 | 47.2 | 55.2 | 42.3 | 47.2 | 55.2 | 42.3 | Not Significant |
| InstructBLIP-13B | 47.2 | 55.8 | 43.5 | 47.2 | 54.9 | 41.7 | 47.2 | 55.1 | 41.8 | Not Significant |
| mBLIP-BLOOMZ | 52.8 | 58.6 | 47.7 | 52.8 | 59.9 | 48.4 | 52.8 | 57.9 | 47.1 | Not Significant |
| GLaMM | 57.8 | 36.9 | 24.0 | 46.4 | 62.8 | 51.1 | 50.0 | 60.4 | 48.9 | Egocentric Relative |
| LLaVA-1.5-7B | 57.7 | 34.7 | 21.4 | 43.5 | 61.5 | 49.8 | 53.3 | 55.8 | 44.4 | Egocentric Relative |
| LLaVA-1.5-13B | 66.6 | 33.5 | 20.8 | 46.5 | 57.7 | 45.7 | 51.3 | 55.1 | 43.1 | Egocentric Relative |
| XComposer2 | 92.6 | 21.9 | 15.8 | 50.5 | 64.4 | 54.3 | 53.8 | 61.9 | 51.4 | Egocentric Relative |
| MiniCPM-V | 77.2 | 33.7 | 26.7 | 52.5 | 62.1 | 51.5 | 53.3 | 61.6 | 51.3 | Egocentric Relative |
| GPT-4o | 73.7 | 42.0 | 35.1 | 55.2 | 60.7 | 50.9 | 54.8 | 60.8 | 51.3 | Egocentric Relative |

Table 8: The full results for testing the preferred frame of reference in VLMs.

| | Back Egocentric | | | Back Intrinsic | | | Back Addressee | | | Front Egocentric | | | Front Intrinsic | | | Front Addressee | | |
|---|---|---|---|---|---|---|---|---|---|---|---|---|---|---|---|---|---|---|
| | Acc% | $\varepsilon^{hemi}_{\times10^2}$ | $\varepsilon^{cos}_{\times10^2}$ | Acc% | $\varepsilon^{hemi}_{\times10^2}$ | $\varepsilon^{cos}_{\times10^2}$ | Acc% | $\varepsilon^{hemi}_{\times10^2}$ | $\varepsilon^{cos}_{\times10^2}$ | Acc% | $\varepsilon^{hemi}_{\times10^2}$ | $\varepsilon^{cos}_{\times10^2}$ | Acc% | $\varepsilon^{hemi}_{\times10^2}$ | $\varepsilon^{cos}_{\times10^2}$ | Acc% | $\varepsilon^{hemi}_{\times10^2}$ | $\varepsilon^{cos}_{\times10^2}$ |
| InstructBLIP-7B | 47.2 | 56.4 | 45.1 | 47.2 | 54.3 | 41.2 | 47.2 | 56.0 | 42.8 | 47.2 | 56.2 | 42.0 | 47.2 | 56.4 | 43.6 | 47.2 | 56.3 | 43.3 |
| InstructBLIP-13B | 47.2 | 49.2 | 38.1 | 47.2 | 54.0 | 40.4 | 47.2 | 53.8 | 40.9 | 47.2 | 63.0 | 49.8 | 47.2 | 58.6 | 46.2 | 47.2 | 59.7 | 47.6 |
| mBLIP-BLOOMZ | 52.9 | 65.4 | 55.3 | 52.4 | 64.7 | 54.8 | 50.8 | 66.3 | 57.1 | 52.6 | 66.2 | 56.3 | 52.1 | 63.8 | 52.8 | 53.1 | 67.1 | 58.9 |
| GLaMM | 47.2 | 46.2 | 32.7 | 47.2 | 54.8 | 42.4 | 47.2 | 62.6 | 49.9 | 47.2 | 40.4 | 25.3 | 47.2 | 55.1 | 41.6 | 47.2 | 51.0 | 38.3 |
| LLaVA-1.5-7B | 49.0 | 41.6 | 27.6 | 47.4 | 56.3 | 45.7 | 46.2 | 66.7 | 55.0 | 47.5 | 39.4 | 25.2 | 47.4 | 52.9 | 39.8 | 47.2 | 41.1 | 27.5 |
| LLaVA-1.5-13B | 47.2 | 38.3 | 22.4 | 47.2 | 53.2 | 41.1 | 47.2 | 52.1 | 39.9 | 47.2 | 48.8 | 36.8 | 47.2 | 54.7 | 41.2 | 47.2 | 41.9 | 26.8 |
| XComposer2 | 65.4 | 40.6 | 26.0 | 52.2 | 57.9 | 47.0 | 54.0 | 58.5 | 47.5 | 86.9 | 27.0 | 17.1 | 52.1 | 58.9 | 47.8 | 53.1 | 57.8 | 46.6 |
| MiniCPM-V | 55.7 | 48.6 | 36.0 | 46.9 | 57.7 | 45.9 | 53.8 | 47.4 | 36.3 | 54.7 | 45.9 | 35.0 | 52.2 | 58.4 | 45.9 | 52.2 | 58.5 | 46.9 |
| GPT-4o | 69.0 | 41.3 | 28.7 | 59.7 | 50.0 | 37.7 | 56.4 | 48.7 | 36.0 | 58.6 | 52.5 | 40.3 | 52.1 | 57.4 | 45.1 | 48.3 | 60.0 | 46.9 |

| | Left Egocentric | | | Left Intrinsic | | | Left Addressee | | | Right Egocentric | | | Right Intrinsic | | | Right Addressee | | |
|---|---|---|---|---|---|---|---|---|---|---|---|---|---|---|---|---|---|---|
| | Acc% | $\varepsilon^{hemi}_{\times10^2}$ | $\varepsilon^{cos}_{\times10^2}$ | Acc% | $\varepsilon^{hemi}_{\times10^2}$ | $\varepsilon^{cos}_{\times10^2}$ | Acc% | $\varepsilon^{hemi}_{\times10^2}$ | $\varepsilon^{cos}_{\times10^2}$ | Acc% | $\varepsilon^{hemi}_{\times10^2}$ | $\varepsilon^{cos}_{\times10^2}$ | Acc% | $\varepsilon^{hemi}_{\times10^2}$ | $\varepsilon^{cos}_{\times10^2}$ | Acc% | $\varepsilon^{hemi}_{\times10^2}$ | $\varepsilon^{cos}_{\times10^2}$ |
| InstructBLIP-7B | 47.2 | 56.3 | 43.3 | 47.2 | 56.0 | 43.0 | 47.2 | 57.9 | 47.1 | 47.2 | 56.8 | 43.5 | 47.2 | 52.9 | 41.5 | 47.2 | 54.4 | 41.0 |
| InstructBLIP-13B | 47.2 | 58.0 | 46.2 | 47.2 | 61.7 | 48.7 | 47.2 | 46.5 | 33.8 | 47.2 | 53.5 | 41.1 | 47.2 | 49.8 | 37.6 | 47.2 | 62.6 | 49.4 |
| mBLIP-BLOOMZ | 51.4 | 65.6 | 55.4 | 46.4 | 67.0 | 56.4 | 47.2 | 64.6 | 54.8 | 50.7 | 65.3 | 54.4 | 48.2 | 63.5 | 52.8 | 47.2 | 62.9 | 52.3 |
| GLaMM | 47.2 | 29.6 | 16.9 | 47.2 | 57.7 | 45.8 | 47.2 | 53.7 | 41.5 | 47.2 | 34.3 | 18.3 | 47.2 | 58.7 | 47.1 | 47.2 | 53.2 | 41.6 |
| LLaVA-1.5-7B | 64.9 | 23.7 | 12.1 | 50.4 | 60.2 | 48.9 | 49.9 | 56.3 | 45.3 | 59.3 | 25.2 | 8.7 | 47.9 | 59.8 | 48.5 | 49.6 | 56.7 | 45.7 |
| LLaVA-1.5-13B | 47.2 | 29.2 | 18.3 | 47.2 | 59.5 | 47.0 | 47.2 | 53.5 | 41.2 | 64.7 | 32.1 | 17.9 | 47.4 | 61.6 | 50.7 | 48.5 | 58.6 | 47.8 |
| XComposer2 | 95.6 | 18.4 | 16.4 | 49.7 | 64.8 | 54.5 | 54.0 | 62.4 | 51.3 | 94.4 | 19.2 | 15.7 | 49.9 | 64.9 | 54.8 | 51.8 | 64.3 | 53.8 |
| MiniCPM-V | 89.4 | 24.2 | 13.3 | 50.4 | 60.9 | 50.0 | 52.1 | 60.6 | 49.9 | 89.9 | 24.3 | 14.2 | 50.1 | 60.6 | 49.5 | 53.3 | 58.0 | 47.3 |
| GPT-4o | 91.7 | 24.0 | 22.8 | 52.5 | 60.1 | 48.6 | 46.7 | 59.9 | 47.6 | 93.9 | 22.1 | 20.5 | 49.2 | 59.4 | 47.0 | 45.1 | 61.0 | 49.1 |

Table 9: The full results for benchmarking perspective-taking performance in VLMs.

| Code | Language | Intrinsic | Egocentric | | | Addressee | | | Code | Language | Intrinsic | Egocentric | | | Addressee | | |
|---|---|---|---|---|---|---|---|---|---|---|---|---|---|---|---|---|---|
| | | | Ref. | Rot. | Tran. | Ref. | Rot. | Tran. | | | | Ref. | Rot. | Tran. | Ref. | Rot. | Tran. |
| af | Afrikaans | 50.9 | 33.7 | 57.8 | 49.2 | 56.6 | 55.5 | 57.5 | ku | Kurdish | 56.5 | 49.5 | 54.4 | 53.1 | 53.1 | 55.2 | 54.0 |
| ak | Akan | 51.8 | 39.6 | 52.2 | 48.8 | 50.4 | 50.6 | 53.8 | la | Latin | 52.2 | 43.9 | 49.8 | 55.5 | 55.1 | 50.8 | 56.6 |
| am | Amharic | 52.1 | 47.4 | 60.7 | 50.9 | 56.8 | 54.2 | 57.6 | lb | Luxembourgish | 54.7 | 35.6 | 57.6 | 50.3 | 58.6 | 53.0 | 59.9 |
| ar | Arabic | 55.7 | 35.8 | 59.0 | 51.0 | 56.6 | 55.8 | 59.8 | ln | Lingala | 52.6 | 45.7 | 50.3 | 59.4 | 54.6 | 51.3 | 58.0 |
| as | Assamese | 51.6 | 40.8 | 55.3 | 51.6 | 48.8 | 52.8 | 56.0 | lo | Lao | 55.8 | 40.1 | 55.0 | 53.7 | 54.7 | 55.7 | 55.5 |
| az | Azerbaijani | 49.8 | 41.9 | 56.2 | 51.6 | 50.7 | 52.4 | 55.5 | lt | Lithuanian | 54.4 | 35.5 | 58.3 | 51.6 | 57.4 | 56.5 | 59.1 |
| be | Belarusian | 54.4 | 39.7 | 61.7 | 46.5 | 51.1 | 51.9 | 58.9 | lv | Latvian | 55.8 | 35.5 | 57.7 | 53.8 | 57.9 | 58.7 | 58.8 |
| bg | Bulgarian | 56.9 | 32.8 | 56.0 | 51.4 | 55.7 | 55.6 | 58.9 | mai | Maithili | 52.7 | 39.8 | 55.1 | 49.9 | 51.2 | 50.6 | 56.5 |
| bho | Bhojpuri | 51.5 | 42.8 | 58.3 | 47.5 | 52.3 | 51.4 | 56.5 | mg | Malagasy | 47.1 | 37.1 | 52.2 | 48.1 | 53.1 | 50.9 | 53.5 |
| bn | Bengali | 55.8 | 34.5 | 57.1 | 50.2 | 53.8 | 56.1 | 57.4 | mi | Maori | 52.0 | 36.6 | 58.5 | 47.6 | 52.0 | 51.9 | 58.1 |
| bs | Bosnian | 55.1 | 35.2 | 58.5 | 49.6 | 54.3 | 53.1 | 59.4 | mk | Macedonian | 54.8 | 37.0 | 59.1 | 49.5 | 56.5 | 56.0 | 58.1 |
| ca | Catalan | 55.6 | 34.7 | 56.9 | 53.0 | 56.0 | 56.4 | 59.7 | mn | Mongolian | 54.1 | 36.7 | 56.8 | 47.4 | 54.7 | 53.6 | 54.7 |
| ceb | Cebuano | 52.9 | 40.0 | 52.7 | 54.9 | 55.3 | 49.5 | 60.3 | mr | Marathi | 52.9 | 34.5 | 55.0 | 48.3 | 51.7 | 52.4 | 55.4 |
| ckb | Sorani | 50.4 | 36.1 | 53.3 | 50.6 | 50.3 | 52.0 | 56.3 | ms | Malay | 54.2 | 33.1 | 55.8 | 50.5 | 55.9 | 55.4 | 57.9 |
| co | Corsican | 57.6 | 35.4 | 57.4 | 54.3 | 58.8 | 58.8 | 59.7 | mt | Maltese | 53.4 | 37.5 | 56.1 | 49.2 | 50.8 | 53.8 | 55.5 |
| cs | Czech | 56.4 | 35.7 | 58.2 | 52.4 | 57.0 | 58.5 | 58.0 | my | Myanmar | 54.9 | 39.3 | 58.7 | 51.8 | 54.2 | 56.1 | 58.1 |
| cy | Welsh | 55.1 | 36.7 | 59.5 | 48.7 | 54.9 | 54.8 | 58.5 | nb | Norwegian | 55.1 | 34.7 | 57.0 | 52.1 | 58.1 | 57.6 | 57.3 |
| da | Danish | 54.9 | 33.0 | 55.2 | 53.1 | 57.6 | 58.3 | 56.9 | ne | Nepali | 53.1 | 39.4 | 58.4 | 47.3 | 52.9 | 54.1 | 54.0 |
| de | German | 55.7 | 36.2 | 58.4 | 52.8 | 56.9 | 56.6 | 60.3 | nl | Dutch | 51.7 | 34.5 | 56.3 | 48.3 | 53.5 | 51.6 | 57.8 |
| el | Greek | 54.4 | 34.4 | 57.1 | 52.2 | 57.1 | 57.5 | 57.7 | nso | Sepedi | 53.8 | 42.6 | 51.0 | 57.1 | 46.3 | 54.2 | 53.7 |
| en | English | 50.9 | 35.8 | 57.3 | 53.7 | 58.8 | 51.3 | 58.8 | ny | Nyanja | 53.7 | 34.5 | 56.6 | 48.0 | 54.4 | 52.8 | 56.7 |
| eo | Esperanto | 58.0 | 34.3 | 56.4 | 54.6 | 58.2 | 58.2 | 60.2 | om | Oromo | 51.1 | 43.5 | 57.3 | 50.6 | 54.9 | 54.8 | 52.9 |
| es | Spanish | 56.9 | 36.2 | 58.1 | 53.3 | 57.0 | 58.5 | 59.0 | pl | Polish | 55.8 | 32.9 | 55.8 | 52.5 | 55.1 | 55.1 | 59.5 |
| et | Estonian | 53.7 | 35.1 | 56.0 | 51.7 | 55.0 | 54.8 | 58.5 | ps | Pashto | 53.0 | 34.6 | 57.4 | 48.9 | 53.7 | 54.6 | 57.4 |
| eu | Basque | 56.8 | 34.3 | 56.8 | 53.2 | 56.7 | 57.1 | 59.5 | pt | Portuguese | 56.3 | 35.9 | 58.2 | 51.9 | 59.1 | 59.3 | 57.6 |
| fa | Persian | 55.8 | 32.1 | 55.3 | 49.8 | 54.4 | 53.8 | 58.0 | ro | Romanian | 57.1 | 34.8 | 57.0 | 53.8 | 58.2 | 58.6 | 59.1 |
| fi | Finnish | 53.9 | 33.7 | 56.7 | 50.8 | 56.3 | 56.1 | 57.6 | ru | Russian | 56.2 | 36.9 | 58.8 | 53.0 | 56.8 | 56.3 | 60.8 |
| fil | Filipino | 50.9 | 31.1 | 54.1 | 49.2 | 54.3 | 54.0 | 55.7 | rw | Kinyarwanda | 53.2 | 35.2 | 56.7 | 48.9 | 54.4 | 54.1 | 57.2 |
| fr | French | 58.0 | 35.2 | 57.4 | 53.7 | 58.6 | 58.5 | 59.4 | sa | Sanskrit | 51.9 | 41.2 | 54.1 | 51.9 | 51.7 | 56.4 | 51.6 |
| fy | Frisian | 53.9 | 38.2 | 58.9 | 49.6 | 53.4 | 53.2 | 59.2 | sd | Sindhi | 51.3 | 40.3 | 56.5 | 49.1 | 54.8 | 49.8 | 57.4 |
| ga | Irish | 54.0 | 33.2 | 55.3 | 49.2 | 52.7 | 51.5 | 53.9 | si | Sinhala | 52.4 | 38.4 | 54.6 | 48.6 | 53.4 | 51.5 | 56.6 |
| gd | Scots Gaelic | 53.9 | 35.4 | 58.5 | 49.6 | 54.7 | 55.8 | 58.1 | sk | Slovak | 56.1 | 37.7 | 57.1 | 54.7 | 57.2 | 56.7 | 59.8 |
| gl | Galician | 56.6 | 37.1 | 59.0 | 53.4 | 57.9 | 57.9 | 60.0 | sl | Slovenian | 55.8 | 36.5 | 59.3 | 49.5 | 53.9 | 54.3 | 58.9 |
| gom | Konkani | 51.1 | 53.1 | 55.5 | 50.5 | 52.5 | 54.9 | 51.8 | sn | Shona | 56.0 | 34.7 | 56.0 | 52.2 | 54.8 | 55.6 | 58.5 |
| gu | Gujarati | 52.6 | 36.6 | 54.2 | 50.9 | 55.5 | 55.3 | 53.8 | so | Somali | 53.7 | 34.3 | 56.4 | 48.2 | 50.0 | 51.6 | 58.4 |
| ha | Hausa | 54.0 | 41.0 | 56.1 | 53.0 | 52.8 | 55.3 | 56.1 | sq | Albanian | 53.6 | 35.1 | 56.4 | 49.0 | 52.6 | 50.3 | 60.1 |
| haw | Hawaiian | 55.3 | 42.2 | 62.1 | 51.5 | 60.5 | 56.2 | 60.8 | sr | Serbian | 55.4 | 34.5 | 57.2 | 50.9 | 52.5 | 55.0 | 58.8 |
| he | Hebrew | 56.5 | 36.4 | 58.8 | 52.5 | 57.1 | 56.5 | 60.3 | st | Sesotho | 53.9 | 38.4 | 55.4 | 51.0 | 51.3 | 54.4 | 55.8 |
| hi | Hindi | 52.5 | 37.8 | 56.6 | 49.1 | 54.5 | 54.6 | 54.2 | su | Sundanese | 51.3 | 36.7 | 55.0 | 50.0 | 53.7 | 50.4 | 57.7 |
| ht | Haitian Creole | 56.1 | 36.0 | 58.3 | 53.6 | 58.4 | 58.2 | 59.6 | sv | Swedish | 54.0 | 33.5 | 56.7 | 51.7 | 55.8 | 56.3 | 58.2 |
| hu | Hungarian | 55.2 | 35.0 | 57.5 | 50.7 | 56.1 | 56.8 | 56.7 | sw | Swahili | 55.3 | 34.2 | 56.8 | 52.4 | 57.2 | 56.5 | 58.4 |
| hy | Armenian | 52.2 | 35.4 | 56.7 | 48.8 | 53.6 | 52.5 | 57.2 | ta | Tamil | 52.0 | 40.4 | 55.2 | 51.1 | 52.2 | 52.9 | 54.6 |
| id | Indonesian | 55.9 | 35.6 | 58.1 | 52.2 | 57.1 | 57.8 | 58.1 | tg | Tajik | 55.7 | 36.7 | 57.7 | 49.7 | 55.4 | 56.6 | 56.3 |
| ig | Igbo | 54.5 | 33.8 | 56.7 | 47.4 | 53.6 | 53.2 | 55.9 | th | Thai | 55.5 | 35.4 | 57.9 | 50.8 | 56.2 | 57.8 | 57.3 |
| ilo | Ilocano | 50.8 | 44.6 | 46.7 | 58.9 | 48.9 | 57.0 | 48.7 | tk | Turkmen | 52.3 | 45.3 | 59.0 | 51.5 | 52.6 | 51.2 | 59.1 |
| is | Icelandic | 55.9 | 34.2 | 57.0 | 52.2 | 56.5 | 58.0 | 57.3 | tr | Turkish | 55.3 | 33.6 | 56.3 | 50.8 | 56.2 | 57.0 | 56.3 |
| it | Italian | 56.8 | 35.6 | 57.6 | 53.6 | 57.9 | 58.2 | 59.6 | ts | Tsonga | 49.4 | 44.6 | 50.0 | 53.7 | 53.3 | 51.5 | 53.6 |
| ja | Japanese | 54.7 | 34.5 | 56.9 | 50.4 | 54.4 | 55.9 | 57.3 | uk | Ukrainian | 56.6 | 36.1 | 58.8 | 50.1 | 56.8 | 55.7 | 59.7 |
| jv | Javanese | 53.5 | 35.3 | 57.7 | 51.0 | 55.7 | 54.8 | 58.9 | ur | Urdu | 52.3 | 34.6 | 56.7 | 49.7 | 54.0 | 55.1 | 57.0 |
| ka | Georgian | 51.1 | 34.8 | 54.3 | 50.6 | 52.0 | 54.0 | 55.3 | uz | Uzbek | 52.6 | 34.5 | 56.4 | 48.1 | 51.7 | 53.2 | 56.6 |
| kk | Kazakh | 52.6 | 36.5 | 58.8 | 50.4 | 54.0 | 56.1 | 56.9 | vi | Vietnamese | 53.9 | 34.6 | 58.5 | 48.6 | 55.7 | 56.1 | 56.8 |
| km | Khmer | 55.6 | 37.6 | 60.2 | 50.5 | 56.9 | 56.6 | 59.3 | yi | Yiddish | 56.7 | 36.5 | 57.9 | 53.5 | 56.8 | 57.5 | 60.0 |
| kn | Kannada | 52.3 | 40.8 | 53.2 | 49.5 | 49.6 | 51.9 | 53.4 | zh | Yoruba | 54.6 | 35.5 | 58.3 | 51.4 | 56.9 | 57.2 | 57.8 |
| ko | Korean | 53.6 | 36.5 | 59.1 | 49.7 | 53.3 | 53.5 | 59.8 | zu | Chinese | 55.6 | 35.9 | 57.7 | 53.1 | 55.3 | 56.5 | 60.2 |
| kri | Krio | 58.3 | 36.2 | 57.1 | 51.2 | 56.1 | 53.2 | 60.2 | | | | | | | | | |

Table 10: The full results for the cross-lingual and cross-cultural evaluation of the preferred frame of reference in VLMs.

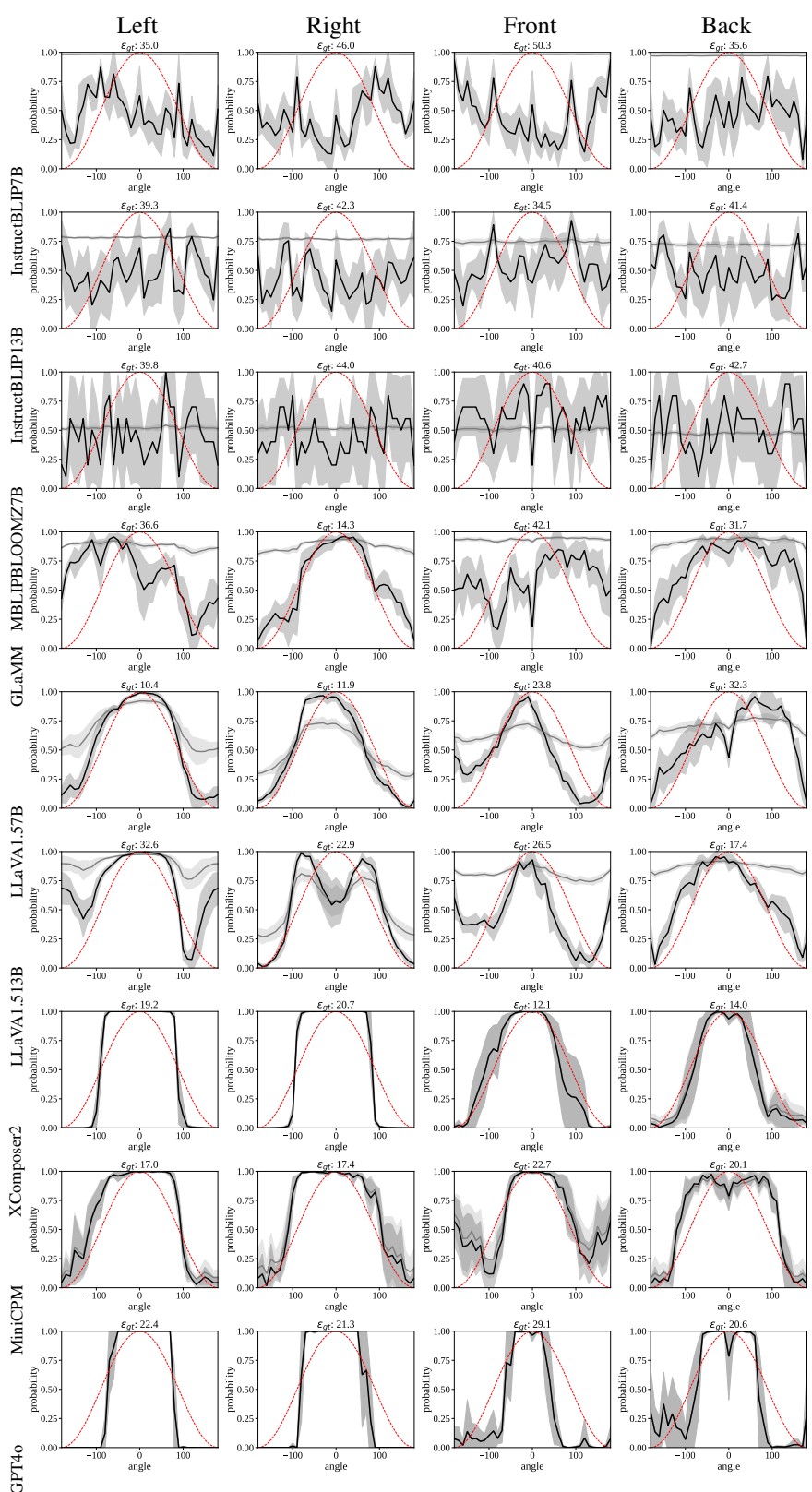

Figure 13: All prediction plots for each model on COMFORT-BALL using the camera perspective prompt (cam). The raw probability $p(\theta)$ in gray, normalized probability $\widehat{p}(\theta)$ in black, and the reference probability $p_{\cos}(\theta)$ of cam in red.

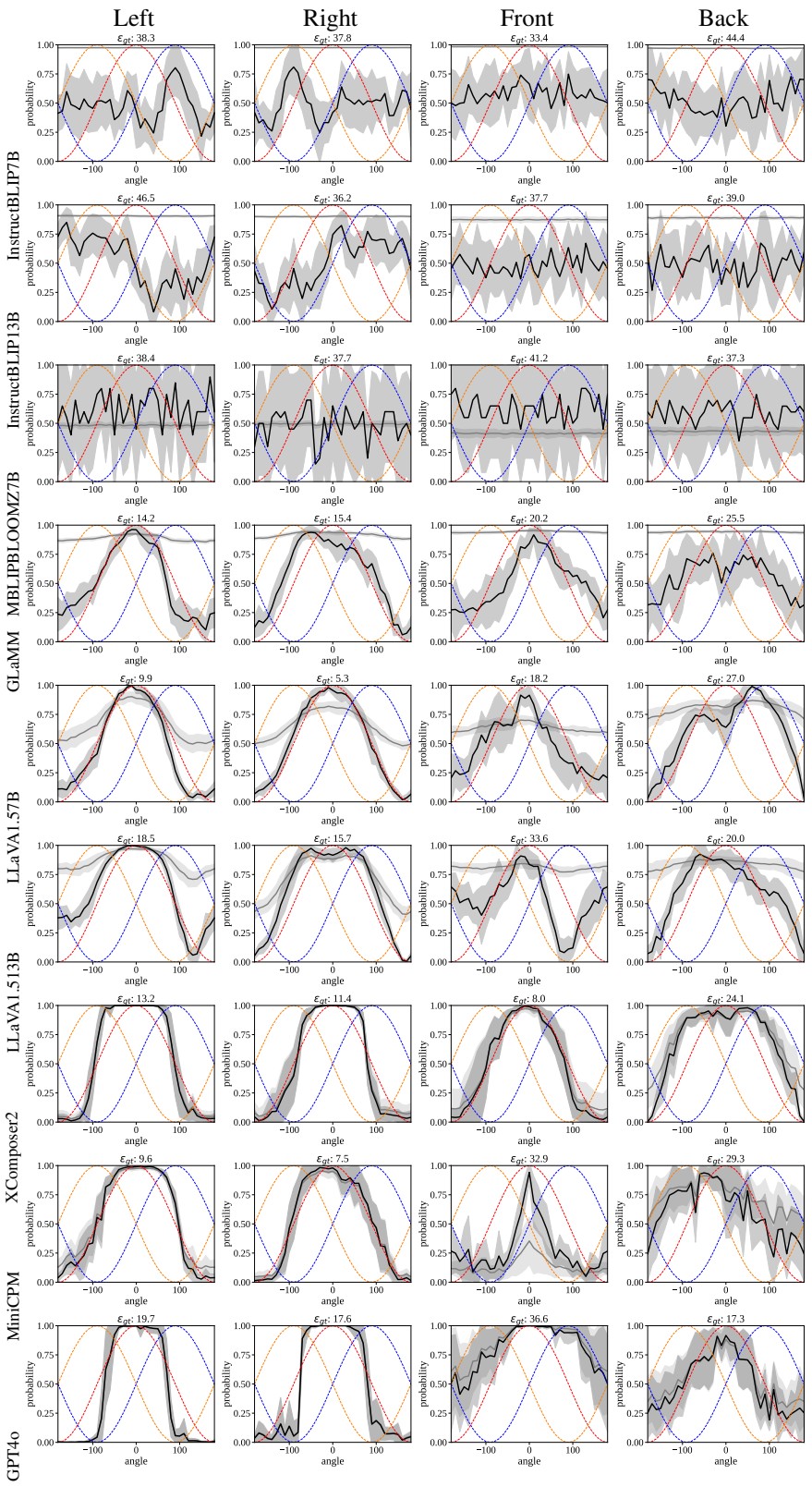

Figure 14: All prediction plots for each model on COMFORT-CAR using the camera perspective prompt (cam). The raw probability $p(\theta)$ in gray, normalized probability $\widehat{p}(\theta)$ in black, and the reference probabilities $p_{\cos}(\theta)$ of cam in red, add in orange, rel in blue. To avoid overlapping reference probabilities of add and rel, we use plots on COMFORT-CAR with relatum facing left for left and right relations and COMFORT-CAR with relatum facing right for front and behind relations.

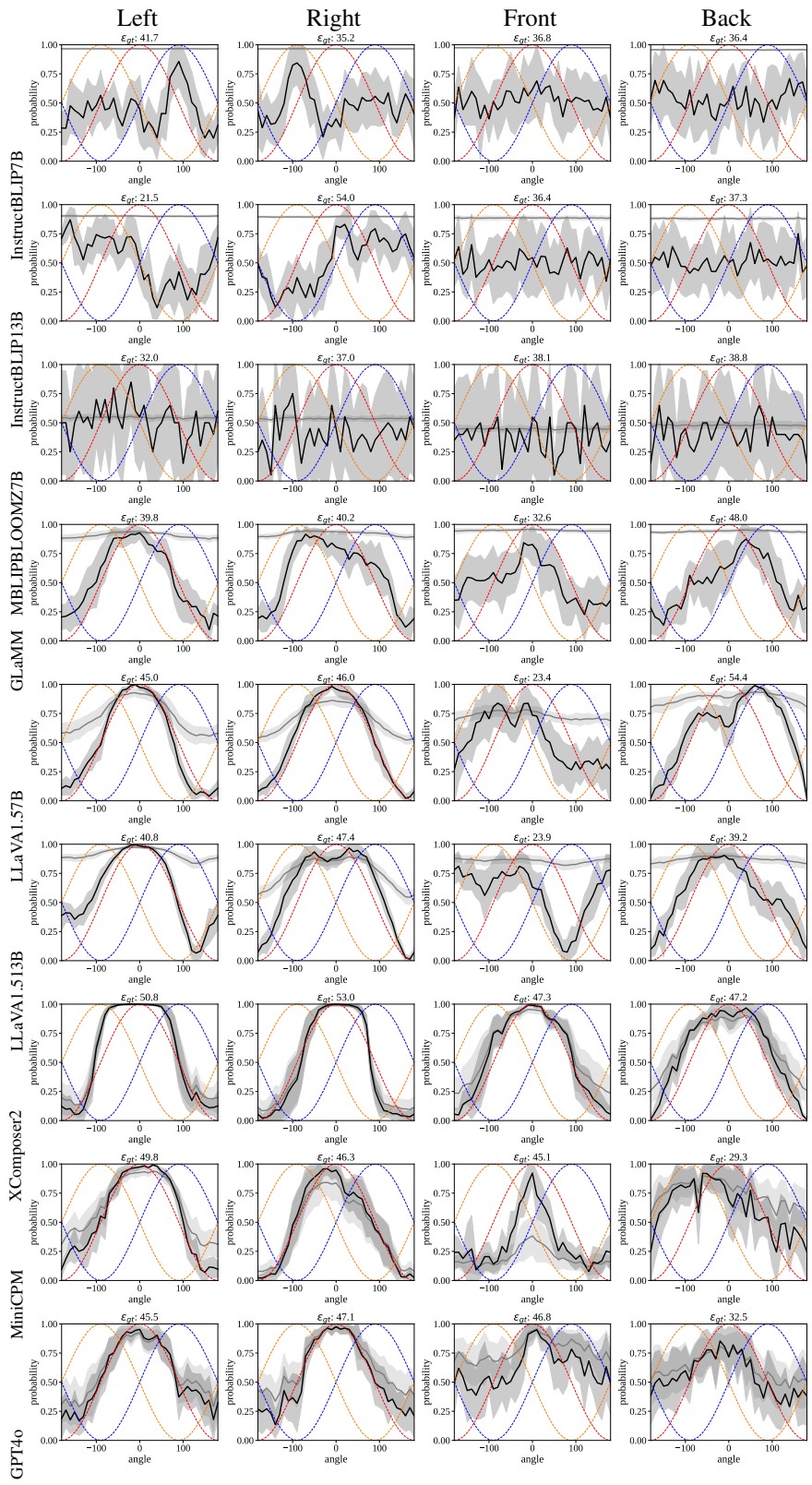

Figure 15: All prediction plots for each model on COMFORT-CAR using the addressee perspective prompt (add). The raw probability $p(\theta)$ in gray, normalized probability $\widehat{p}(\theta)$ in black, and the reference probabilities $p_{\cos}(\theta)$ of cam in red, add in orange, rel in blue. To avoid overlapping reference probabilities of add and rel, we use plots on COMFORT-CAR with relatum facing left for left and right relations and COMFORT-CAR with relatum facing right for front and behind relations.

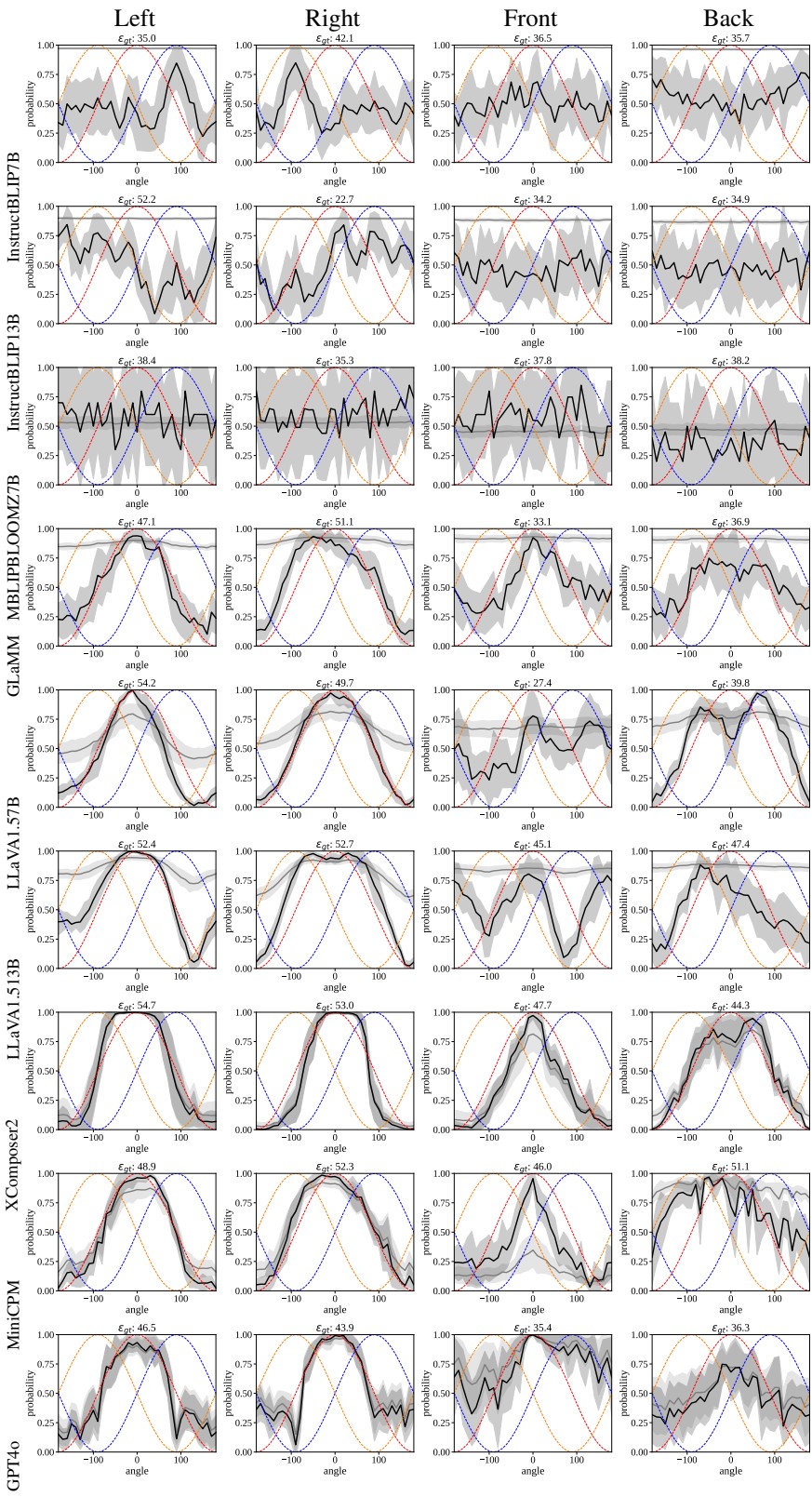

Figure 16: All prediction plots for each model on COMFORT-CAR using the relatum perspective prompt (rel). The raw probability $p(\theta)$ in gray, normalized probability $\hat{p}(\theta)$ in black, and the reference probabilities $p_{\cos}(\theta)$ of cam in red, add in orange, rel in blue. To avoid overlapping reference probabilities of add and rel, we use plots on COMFORT-CAR with relatum facing left for left and right relations and COMFORT-CAR with relatum facing right for front and behind relations.

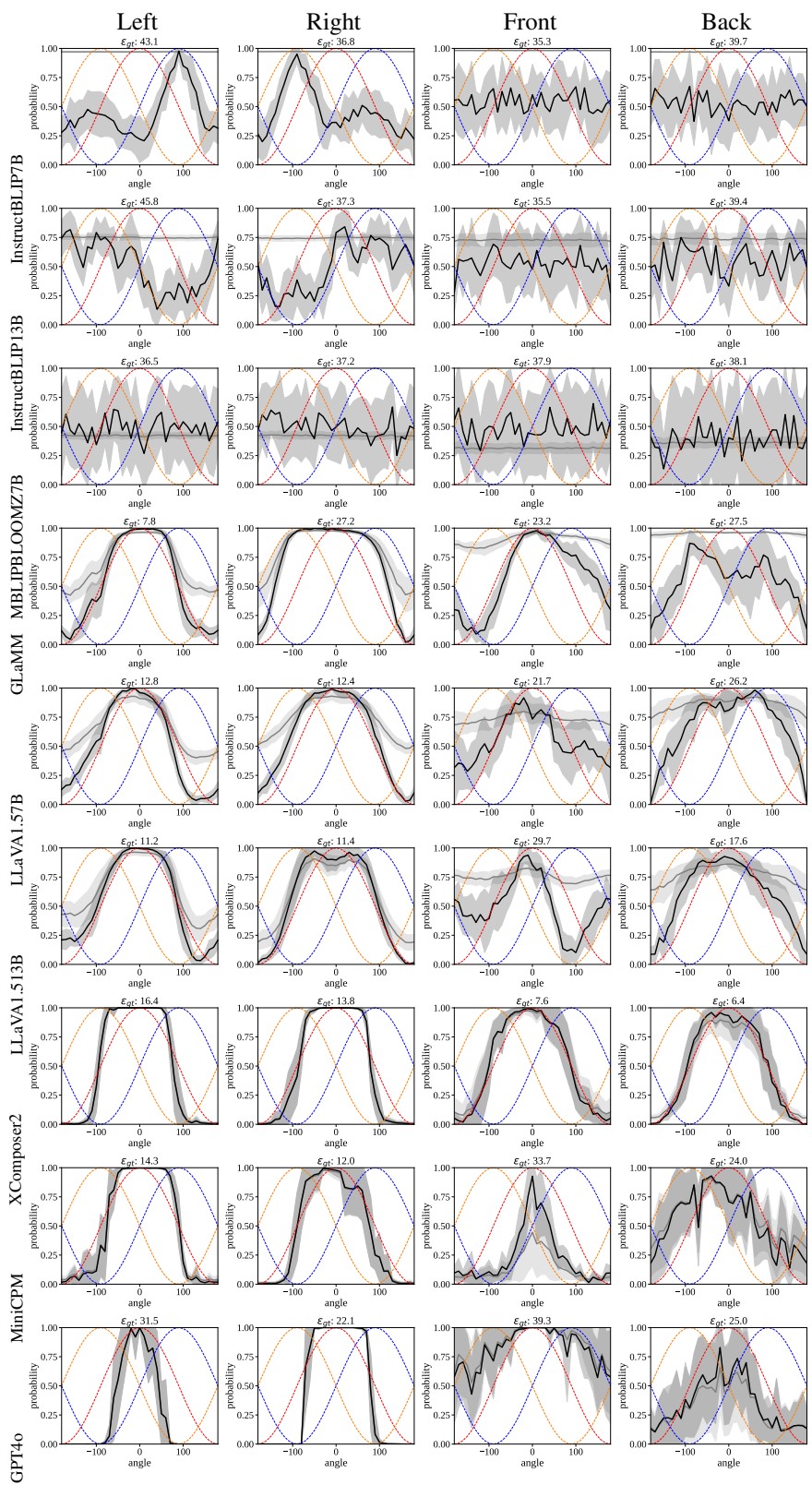

Figure 17: All prediction plots for each model on COMFORT-CAR without perspective prompt (nop). The raw probability $p(\theta)$ in gray, normalized probability $\widehat{p}(\theta)$ in black, and the reference probabilities $p_{\cos}(\theta)$ of cam in red, add in orange, rel in blue. To avoid overlapping reference probabilities of add and rel, we use plots on COMFORT-CAR with relatum facing left for left and right relations and COMFORT-CAR with relatum facing right for front and behind relations.

