# OpenReview forum: "Do Vision-Language Models Represent Space and How? Evaluating Spatial Frame of Reference under Ambiguities"
_ICLR.cc/2025/Conference — ICLR 2025 Oral_

### Official Review · Reviewer_EAzE · 2024-10-19

**Soundness:** 3
**Presentation:** 3
**Contribution:** 3
**Rating:** 5
**Confidence:** 3

**Summary:**

- Investigates how vision-language models (VLMs) represent and reason about spatial relations, particularly when spatial language is ambiguous due to differing frames of reference (FoRs). It focuses on evaluating the robustness, consistency, and flexibility of VLMs in handling such ambiguities.

- The authors introduce a new evaluation protocol called the COnsistent Multilingual Frame Of Reference Test (COMFORT), which systematically assesses VLMs' spatial reasoning capabilities. The evaluation includes tasks with synthetic 3D images and text descriptions of spatial relations, with tests conducted in 109 languages across 170 regions worldwide.

- The study reveals that while VLMs show some alignment with English conventions for resolving spatial ambiguities, they struggle with: (A) Consistency and robustness in spatial reasoning, (B) Flexibility in adopting multiple FoRs, (C) Cross-linguistic and cross-cultural understanding of spatial relations, with English conventions often dominating over those of other languages.

**Strengths:**

- Unlike existing benchmarks, which typically focus on spatial language understanding within a single frame of reference or specific cultural context, this work broadens the scope by emphasizing ambiguity in spatial reasoning due to different Framework of References and cross-linguistic/cross-cultural diversity.

- The paper is well-structured, with clear explanations of key concepts such as frames of reference and spatial reasoning.

More importantly,

- The paper’s contributions are highly significant in the context of both vision-language research and broader discussions on AI alignment with human cognition. By focusing on spatial reasoning, a core component of human cognition, this work addresses a critical gap in the evaluation of VLMs.

- The cross-lingual evaluation across diverse cultural backgrounds provides valuable insights into the limitations of current VLMs in handling non-English spatial conventions. This has implications for the development of more globally applicable models, as the dominance of English-centric FoRs could hinder the usability of VLMs in non-English contexts.

**Weaknesses:**

- While the introduction of the COMFORT benchmark is a novel contribution, the idea of testing spatial reasoning in vision-language models is not entirely new. Several previous benchmarks, such as those mentioned in the paper, have evaluated spatial reasoning capabilities in VLMs. The novelty of this paper lies primarily in its focus on the frame of reference ambiguities, but the paper does not sufficiently clarify how this focus significantly advances beyond the existing benchmarks. But I agree that the frame of reference is a rather novel take on the same topic.

- Although the paper provides valuable insights into how English conventions dominate spatial reasoning in multilingual models, the discussion around cultural and linguistic biases remains relatively high-level.

- The paper identifies that most models tend to default to egocentric relative frames of reference, which aligns with English spatial conventions. However, it does not deeply investigate whether this bias toward egocentrism is inherently problematic across all contexts or if there are scenarios where this behavior is acceptable or even preferable.

+ (not required) It would be beneficial to include a dedicated section discussing the practical relevance of the findings in LLM applications. Is there an area the the findings of this paper can seriously affect?

**Questions:**

NA

---

> ### Author Response · Authors · 2024-11-25
>
> We sincerely thank reviewer EAzE for their insightful feedback and constructive comments, as well as for recognizing the novelty of our benchmark, the clarity of our writing, the significance of our work in vision-language research and AI alignment, and its implications for developing more globally applicable models. **Given the overall positive sentiment of the review, we kindly remind the reviewer that a score of 5 is marginally below the acceptance threshold.** In case there are any misunderstandings or mistakes, we appreciate it if the reviewer could reconsider the rating.
>
> Below, we address the identified weaknesses, questions, and suggestions.
>
> ### Response to Weakness #1 (novelty):
>
> **Thank you for acknowledging that “frame of reference is a rather novel take”!**
>
> As highlighted in the Introduction, frames of reference are crucial for studying spatial cognition across modalities, providing a foundational framework for understanding how spatial relationships are perceived, interpreted, and communicated. Selecting the appropriate FoR is key to resolving ambiguities in spatial language understanding, whether by following explicit instructions through perspective prompts or adhering to language conventions. To the best of our knowledge, existing benchmarks do not account for this aspect, as most assume an egocentric relative FoR with reflected coordinate transformation, aligning with English preferences. However, English speakers can adapt to alternative FoRs when instructed, and speakers of other languages may exhibit vastly different preferences.
>
>
> ### Response to Weakness #2 (Multilingual study remains high level)
> In this experiment, **we hope to provide VLM’s preference and raise concerns that English may dominate the FoR preference conventions of other languages in multilingual VLMs**.
>
> The ground truth human preferences are based on established findings in the linguistics literature, mostly from Levinson, 2003 but also in Majid et al., 2004; O’Meara & Báez, 2011; and Bender et al., 2020. For example, Dutch, English, Japanese, Tamil, Hausa, Spanish, Norwegian, Chinese, Tzeltal, Tongan, and Farsi prefer Relative FoR; and Jaminjung, Mopan, Totonac prefer Intrinsic FoR. In the revision, we explicitly stated this in Section 4.5 and cited the corresponding studies.
>
> Although there is a good number of languages with well-documented frame-of-reference preferences and established consensus in cognitive psychology. However, not all languages have been extensively studied or reached such a consensus. Our ongoing work aims to address this gap through systematic human studies. Unfortunately, this research falls beyond the scope of an AI conference paper, so we leave it for future exploration.
>
> ### Response to Weakness #3 (When is egocentrism problematic)
> > “However, it does not deeply investigate whether this bias toward egocentrism is inherently problematic across all contexts or if there are scenarios where this behavior is acceptable or even preferable.”
>
> While there is no global ground truth, the language we studied in the main paper demonstrates a known preference, e.g., as the reviewer noted, English prefers an egocentric relative FoR. To this end, better alignment with the language convention and intuition is preferred in communication.
>
> Besides, the perspective-taking capability of VLMs (Section 4.3) can function as a benchmark, as the input and ground truth are unambiguous. We encourage future work in VLM training to address this specific challenge.
>
> Therefore, a "bias toward egocentrism" is acceptable when communicating with English speakers by default or when explicitly instructed to do so. However, it is not acceptable when communicating with speakers of languages that prefer other conventions or when models are instructed to adopt an alternative perspective, such as the addressee's viewpoint. **Alignment with human cognition and cultural conventions is preferred.**
>
> ### Response to Weakness #4 (Practical relevance)
>
> Our findings provide valuable insights for VLM developers. As we mentioned in Section 5, “current training recipes for multilingual multimodal language models heavily rely on machine-translated captions (Chen et al., 2023c; Geigle et al., 2024); however, this practice can be problematic…To enable similar linguistic transmission in AI models, exposure to naturally
> generated multilingual image-text data is crucial (Romero et al., 2024).”
>
> Also, VLMs have many applications in embodied AI and robotics and spatial reasoning is critical for making decisions in the 3D world. Moreover, embodied communication requires flexibility for accommodating different FoRs. For example, if you want a robot to pick up the cup on your left using VLMs, the robot has to reason from your perspective to select the cup for manipulation.

---

### Official Review · Reviewer_Jhao · 2024-10-25

**Soundness:** 4
**Presentation:** 3
**Contribution:** 3
**Rating:** 6
**Confidence:** 3

**Summary:**

This paper discusses the performance of VLMs in handling ambiguity in spatial expressions. The authors constructed a new benchmark -COMFORT to systematically test the spatial reasoning capabilities of VLMs, and conducted analysis on COMFORT.

**Strengths:**

1. This paper delves into "Do VLM represent space and how". Specifically, it considers the concept of Frames of Reference (FoR) and constructs a framework for understanding how VLMs perceive spatial information, taking into account the ambiguities that FoR might bring. The COMFORT framework is systematic and comprehensive, and its task formulation is rigorous. The authors create challenging but realistic queries using synthetic 3D images and corresponding textual descriptions, and propose systematic metrics. From the perspective of benchmark construction, this is a systematic and rigorous work.

2. The authors tested different VLMs and analyzed how they internally understand spatial representations. The testing methods are reliable.

3. The authors extended the test cases to cover 109 languages across 170 regions, further testing how VLMs respond to ambiguities in spatial relationship descriptions under different languages.

**Weaknesses:**

As the author claimed, "spatial expressions in situated communication can be ambiguous." Therefore, considering that humans also exhibit different understandings when dealing with similar issues of ambiguity, even if VLMs demonstrate different interpretations, it does not necessarily imply a clear superiority or inferiority. In other words, the evaluation metrics proposed in the paper are not absolute criteria, like egocentric is not naturally better than addressee-centered, but rather serve as perspectives for observing how VLMs understand spatial information.

From this standpoint, I believe the author should include more analysis of how humans understand ambiguities in spatial relations (relevant information can be found in psychological and cognitive science research). This would help us better understand the differences between VLMs’ understanding of these ambiguities and the human perspective, and ultimately facilitate a better alignment of spatial reasoning in vision-language models with human cognition.

Hope to see further discussion on these points.

**Questions:**

Please refer to the comments in the Weakness.

---

> ### Author Response · Authors · 2024-11-25
> **Response to Reviewer Jhao**
>
> We sincerely thank the reviewer for your thoughtful and constructive feedback (rephrase using ChatGPT later), and for recognizing our paper to be systematic and rigorous, having reliable testing methods, and extensive multilingual analysis. Below, we respond to weaknesses.
>
> ### Clarification
> > “In other words, the evaluation metrics proposed in the paper are not absolute criteria, like egocentric is not naturally better than addressee-centered, but rather serve as perspectives for observing how VLMs understand spatial information.”
>
> This is true as we emphasize that this dataset and its associated evaluation metrics are designed to assess cognitive similarity and alignment with human spatial cognition.
>
> However, we also emphasize that:
> - **While there is no global ground truth, the language we studied in the main paper demonstrates a known preference**, e.g., English prefers an egocentric relative FoR. To this end, **better alignment with the language convention and intuition is preferred in communication**.
> - The perspective-taking capability of VLMs (Section 4.3) can function as a benchmark, as the input and ground truth are unambiguous. We encourage future work in VLM training to address this specific challenge.
>
> ### Response to Weakness (Insufficient psychological and cognitive science related work)
> We kindly refer to Section 2.1 for how humans resolve ambiguities based on linguistic and cultural backgrounds, and we use human preference as the criteria for evaluation. For example, English prefers the egocentric relative FoR with reflected coordinate transformation, so when we are evaluating English, we use the reflected coordinate transformation as the ground truth.
>
> These ground truth human preferences are based on established findings in the linguistics literature, mostly from Levinson, 2003 but also in Majid et al., 2004; O’Meara & Báez, 2011; and Bender et al., 2020. For example, Dutch, English, Japanese, Tamil, Hausa, Spanish, Norwegian, Chinese, Tzeltal, Tongan, and Farsi prefer Relative FoR; and Jaminjung, Mopan, Totonac prefer Intrinsic FoR. In the revision, we explicitly stated this in Section 4.5 and cited the corresponding studies.
>
> Although there is a good number of languages with well-documented frame-of-reference preferences and established consensus in cognitive psychology. However, not all languages have been extensively studied or reached such a consensus. Our ongoing work aims to address this gap through systematic human studies. Unfortunately, this research falls beyond the scope of an AI conference paper, so we leave it for future exploration. In this work, we hope to provide VLM’s preference and raise concerns that English may dominate the FoR preference conventions of other languages in multilingual VLMs with this example.

---

> > ### Comment · Reviewer_Jhao · 2024-11-27
> > **Thanks for authors' rebuttal**
> >
> > I read the author's rebuttal as soon as it was posted. After thoroughly reviewing the rebuttal, considering the comments from other reviewers, and revisiting the paper, I have decided to maintain my original positive score.
> >
> > Thank you!

---

### Official Review · Reviewer_GnST · 2024-10-29

**Soundness:** 4
**Presentation:** 3
**Contribution:** 3
**Rating:** 8
**Confidence:** 4

**Summary:**

This paper explores the spatial reasoning capabilities of vision-language models (VLMs) and their handling of spatial ambiguities. It introduces the COnsistent Multilingual Frame Of Reference Test (COMFORT) to systematically evaluate these models. The study reveals that while VLMs show some alignment with English conventions in resolving spatial ambiguities, they exhibit significant shortcomings: poor robustness and consistency, lack of flexibility to accommodate multiple frames of reference (FoRs), and failure to adhere to language-specific or culture-specific conventions in cross-lingual tests. The paper emphasizes the need for more attention to the ambiguous nature and cross-cultural diversity of spatial reasoning in VLMs.

**Strengths:**

This paper introduces the innovative COnsistent Multilingual Frame Of Reference Test (COMFORT), which systematically evaluates vision-language models (VLMs) across multiple languages and cultural contexts. This novel approach highlights the importance of considering linguistic and cultural variations in model performance. The research is thorough and methodical, assessing nine state-of-the-art VLMs with detailed analyses of their robustness, consistency, and flexibility. The paper is well-organized and clearly written, making complex concepts accessible. Its findings underscore the need for more robust and flexible models, providing a valuable framework for future research in vision-language modeling and spatial reasoning.

**Weaknesses:**

The scope of spatial relations is limited, focusing mainly on basic relations like front-back and left-right, while neglecting others such as near-far and above-below. The evaluation relies on synthetic 3D images, which may not fully capture real-world complexities, and lacks consideration for occlusion and varying camera angles. Additionally, the analysis could benefit from a more diverse linguistic and cultural context, as well as human annotations for the multilingual dataset. Addressing these weaknesses by expanding the scope of spatial relations, incorporating real-world scenarios, and improving the quality of the dataset with human annotations would provide a more comprehensive evaluation of VLMs' spatial reasoning capabilities.

**Questions:**

1. How do the findings impact the development of embodied AI systems and other applications like autonomous driving, robotics, or augmented reality?

2. How could improved spatial reasoning in vision-language models (VLMs) enhance performance in multilingual and multicultural contexts?

3. What are the key areas of future research to advance the spatial reasoning capabilities of VLMs, and what specific challenges or limitations need to be addressed?

---

> ### Author Response · Authors · 2024-11-25
> **Response to Reviewer GnST**
>
> We appreciate your thoughtful feedback and valuable insights, and for recognizing our novel approach, thorough and methodical studies, and well-organized and clear writing. Below, we respond to weaknesses, questions, and suggestions.
>
> ### Response to Weakness #1 (Limited scope of spatial relations)
> Our paper focuses on the ambiguous nature of spatial language and builds upon linguistic and cognitive science literature on spatial frames of reference. We chose lateral and sagittal directions as they are the most fundamental spatial indicators.
> > “There is a tight connection between the Relative FoR and the Intrinsic FoR: it seems that you cannot have a Relative FoR without an Intrinsic FoR. Like the Intrinsic FoR, the Relative FoR requires ‘parsing’ of objects – most importantly, a parsing of the self into front, back, left and right.” (Majid et al., 2004)
>
> As we mentioned in the Limitations section, spatial relationships like near-far and above-below are left to future work as we need novel metrics to evaluate these terms.
>
> ### Response to weakness #2 (Synthetic 3D images may not fully capture real-world complexities)
> Our synthetically generated images have occlusions and varying camera angles. We refer to Section 3.2 and Figure 3 for examples of the rendered images. For reasons of using only synthetic images in our dataset:
> It’s hard to rotate the target object around the reference object in the real world with uniform steps for collecting natural images.
> We also want to carefully control other variables (e.g. camera pose) in the experiments for systematic analysis, which are hard to control in capturing natural images.
>
> However, we added a case study to see if results in synthetic images generalize to real photos. We fixed the camera pose and manually rotated the target object around the reference object using (1) two equally sized red and blue balls; and (2) a laptop and the red ball. We keep everything else identical to the original COMFORT setup. We found that our findings still hold for alternative backgrounds. Please refer to Section Appendix C for more details.
>
> ### Response to Weakness #3 (A more diverse linguistic and cultural context or human annotations)
> While we have started this study and formulated the problem, more extensive investigation into more languages and cultures will be an interesting direction for future work. We agree that human annotations for the multilingual dataset will improve the quality, but due to the large number of images and languages we evaluated, it requires a collective effort from the community to make this happen: We need native speakers of different languages from different cultural backgrounds to label in total (720 + 57600) * 109 = 6356880 instances.
>
> In fact, our ongoing work aims to address this gap through systematic human studies. Unfortunately, this research falls beyond the scope of an AI conference paper, so we leave it for future exploration. In this work, we hope to provide VLM’s preference and raise concerns that English may dominate the FoR preference conventions of other languages in multilingual VLMs with this example.
>
> ### Response to Question #1:
> Our findings suggest that current VLMs are not robust and consistent enough so we should be cautious when we apply VLM to autonomous driving. Additionally, spatial reasoning from another person’s perspective is important in embodied communications, but our experiments show that VLMs cannot flexibly accommodate multiple FoRs. Moreover, the failure to adhere to language-specific or culture-specific conventions in cross-lingual tests makes them unusable for people in a non-English speaking country.
>
> ### Response to Question #2:
> As discussed in Section 5, “current training recipes for multilingual multimodal language models heavily rely on machine-translated captions,” which can introduce significant challenges. Enhancing spatial reasoning is a critical step toward the broader goal of achieving better multilingual and multicultural alignment in vision-language reasoning, ultimately contributing to the development of fair AI.
>
> ### Response to Question #3:
> In Section 5 Discussions, we mainly discussed (1) future work is necessary to improve the consistency and robustness of spatial representations in these models, (2) future work should extend the current 2D VLMs to the 3D domain, by considering camera poses and multiview data (Yang et al., 2024) for training, (3) to enable similar linguistic transmission in AI models, exposure to naturally generated multilingual image-text data is crucial.

---

### Official Review · Reviewer_4ug1 · 2024-11-04

**Soundness:** 4
**Presentation:** 4
**Contribution:** 3
**Rating:** 8
**Confidence:** 4

**Summary:**

**Main contributions**:
The paper has two main contributions:
1. It presents COMFORT, a framework for evaluating VLMs' understanding of spatial frame of reference expressions, which relate two objects to one another (e.g. "Is the basketball *to the right of* the car?"). Expressions are evaluated using a rendered image containing the two objects and which is used to query the VLM with (as a binary Yes/No question) for whether or not the relation holds for the image.

2. It uses COMFORT to show that a set of contemporary VLMs have distinct preferences for frame of reference parameters. Specifically, the paper presents evidence for the preference of English language frame of reference parameters in VLMs.

**Framework Structure**:
Expressions in COMFORT vary across two main conditions:
1. Expressions which include two objects with a semantic "front" (e.g. such as the "front" of a person being the side with the person's face). In this scenario the spatial expression must be reasoned about in terms of an anchor frame of reference (i.e. either object A's, object B's, or the camera's frame). The selection of the frame of reference will determine which direction in the image corresponds to "left of", "back of" etc.

2. Expressions which include objects without a semantic "front". In COMFORT, this is instantiated by scenarios with two balls. In such scenarios, a coordinate transformation convention must be assumed for determining what is "right", "left", "back", etc. Different languages have different standard conventions, and so this condition is set up to test this.

Preferences are measured by evaluating the [Yes/No] probability for a given spatial expression in relation to a ground truth *region of acceptability*, a continuous region of rotation of the second object around the center object (see Figure 1c for reference) in which a spatial expression may hold true. This region of acceptability allows for not only for accuracy to be measured but also for the dynamics of the probability to be measured as functions of distance from the center of the region of acceptability -- intuitively, the score of "right of" should decrease smoothly as the object is rotated from 0 to 90 degrees from the center of the region. Specifically, the paper presents.
These measures probe the robustness of model scores (e.g. evaluating std across object variations unrelated to space like color or shape) as well as the consistency (are scores symmetric at both equivalent angles from the acceptability region's center? Do they change smoothly as angle is varied?).

**Results**
The paper uses the region of acceptability based error measures to show that VLMs generally skew towards preferring egocentric frames of references (condition 1) and the reflected coordinate transformation convention. Additionally, the paper also shows that these preferences cannot be reliably changed when the prompt explicitly defines the frame of reference to be used, or when a language (other than English) with different known preferences is used to prompt the model.

**Recommendation**
This is a solid paper which I believe presents a unique contribution that stands to better capacitate the community in vetting the spatial understanding of VLMs. I have noted a few suggestions below under Weaknesses to improve the presentation of the paper. With that said, I think it is already at a good quality and I recommend acceptance.

**Strengths:**

**Strong Backing in Theory**: The paper's framing and setup in sections 1 through 3 (which motivates the problem statement and framework design choices) is a joy to read. Motivating statements and scoping within existing work is very clear and compelling.

**Conclusions Well Supported**: The experimental setup is quite thorough and shows clear trends which the authors use to justify their claims. A majority of models are clearly shown to have FoR preferences based on the metrics *cos* and *hem* error functions defined, with the BLIP family of VLMs being a notable exception. I particularly appreciated that the paper additionally presents a compelling explanation for the BLIP discrepancy by showing that these same models score poorly on an evaluation of object hallucinations.

**Novelty**: In my understanding I believe that the presented framework is novel based on its continuous definition based on regions of acceptability. This seems to me to be an important distinction namely because these spatial relationships are known to lie within continuums and are not discrete.

**Clarity**: The paper's writing is generally clear, and I found the organizational structure helpful for understanding the concepts introduced.

**Weaknesses:**

**Cross-lingual Experiment Presentation**: I think some further refinement/elaboration of the cross lingual experiments (Section 4.5) could help strengthen the paper. Although the paper does state upfront that it spends most of the content focused around English experiments, the treatment of the section still felt a little too short for the emphasis that is placed in the introduction for this being a core piece of the framework (e.g. the M in COMFORT stands for Multilingual). Specifically I think a couple of details could be expanded upon.

First, the results in Figure 8 are a little ambiguously presented. Is the world map mapping the preferences of GPT-4o, or is it mapping ground truth human preferences? I believe the former, but the wording in both the figure and the paper makes it sound like these are ground truth preferences.
Second, for ground truth preferences, I'm wondering how these are obtained? -- I'm assuming this is pulled from the linguistics literature, but it is not explicitly stated and appears like a key detail to me. I would appreciate it if the authors could clarify both of these points.

Lastly, although the results presented clearly show a preference for the same reference frames as English, I think they could be strengthened by also showing a notion of what the expected preferences would be if the VLM were to be adhering to the conventions of the language being used. For example, I think this could be done by (a) adding a second map showing a visualization of ground truth human preferences, or (b) adding the bold/underline convention from the table in Figure 8 to Table 10.

**Questions:**

**Suggestions**:
* L212 "the query is appended after four different perspective prompts...": I'm assuming this meant to say "after *one of* four different..."?

* I understand the authors may have been space limited (no pun intended), but I think adding a Conclusion section to reiterate contributions and takeaways (further than the Discussion section already does) could make the paper stronger. There's a large amount of content, so spoon-feeding the final takeaways to readers could aid clarity.

* This is minor, but relatedly I think the paper could be a little clearer about what the intended usage of the framework is for readers. Are readers supposed to come away with an understanding that this will be a benchmark they can evaluate their own models on? I believe yes, but the wording of the paper at the beginning and end doesn't make this explicit. In the introduction, COMFORT is introduced as a "framework" and not a "benchmark", was this intentional? Secondly, the paper closes with only a discussion over the empirical findings in Section 5, without any explicit reiteration that it has introduced COMFORT. This omission at the end made it feel like the only point of the paper was to present a study of VLM spatial understanding -- this would be fine of course, but if I'm understanding correctly I think the paper would additionally want to explicitly market itself as offering COMFORT as a testbed for VLMs. It's subtle, but I think being explicit about this, especially at the end of the paper, could really help drive home the point the authors want to communicate.

---

> ### Author Response · Authors · 2024-11-25
> **Response to Reviewer 4ug1**
>
> We are grateful for your thorough evaluation and positive feedback on our motivations from linguistics and cognitive science, well-supported conclusions and novel continuous definitions based on regions of acceptability, and clear writing structure. Below, we respond to weaknesses, questions, and suggestions. The main weakness identified by the reviewer is the refinement/elaboration of the cross-lingual experiments, which we detailed below:
>
> ### Response to Weakness 1 (the results in Figure 8 are a little ambiguously presented):
> We confirm that Figure 8 reflects GPT-4o's preferences based on the cosine parsing error, weighted by the speaking population of the top three languages in each region. To eliminate ambiguity, we revised the figure caption and the associated text in Section 4.5 to explicitly state this.
>
>
> ### Response to Weakness 2 (ground truth preferences and conventions):
> Ground truth human preferences are based on established findings in the linguistics literature, mostly from Levinson, 2003 but also in Majid et al., 2004; O’Meara & Báez, 2011; and Bender et al., 2020. For example, Dutch, English, Japanese, Tamil, Hausa, Spanish, Norwegian, Chinese, Tzeltal, Tongan, and Farsi prefer Relative FoR; and Jaminjung, Mopan, Totonac prefer Intrinsic FoR. In the revision, we explicitly stated this in Section 4.5 and cited the corresponding studies.
>
> Although there is a good number of languages with well-documented frame-of-reference preferences and established consensus in cognitive psychology. However, not all languages have been extensively studied or reached such a consensus. Our ongoing work aims to address this gap through systematic human studies. Unfortunately, this research falls beyond the scope of an AI conference paper, so we leave it for future exploration. In this work, we hope to provide VLM’s preference and raise concerns that English may dominate the FoR preference conventions of other languages in multilingual VLMs with this example.
>
> ### Response to Suggestions (Intended usage of the framework)
> We intentionally use the term “framework” to emphasize that this dataset and its associated evaluation metrics are designed to assess cognitive similarity and alignment with human spatial cognition. While each studied language demonstrates a preference, we do not position this as a leaderboard-driven benchmark. However, the perspective-taking capability of VLMs (Section 4.3) can function as a benchmark, as the input and ground truth are unambiguous. We encourage future work in VLM training to address this specific challenge.

---

> > ### Comment · Reviewer_4ug1 · 2024-11-28
> >
> > Thanks to the authors for their thorough response to my questions and concerns. I maintain my original assessment and score. I believe that this is a solid paper and will be of interest to the community -- I recommend acceptance to the conference.

---

### Official Review · Reviewer_QwFz · 2024-11-08

**Soundness:** 4
**Presentation:** 4
**Contribution:** 4
**Rating:** 10
**Confidence:** 4

**Summary:**

The authors assess how VLMs represent space through the lens of “situated communication,” framing the meaning of spatial relations like “to the right of” under transformations, in different frames of reference, and others (situated communication).

They introduce the COMFORT dataset, tasks, and metrics to analyze spatial reasoning under these classes of transformations, and see which frame-of-reference preferences LMs have, and test whether these preferences are language-agnostic by extending their evaluation to multilingual settings.

The images are rendered to enable simple dynamic generation of test samples. Their language queries that test the relations likewise have a simple structure and are programmatically produced, enabling translation into other languages. Testing the kinds of relations is simple using this methodology and allows them to test for continua such as how a model rates the “in front of” relation by angle between the two in the plane relative to the camera.

**Edit**: The authors did a great job adding some small case studies that **show** how their results generalize to other backgrounds and real images (although these are still pretty close to in-distribution to the test images). The use of a laptop rather than a ball in the real images in particular is a strong change. I have raised all the component scores to 4. **Although I would prefer an option to give a 9/10, since it isn't available I will raise from 8 to 10.**

These tests are then translated into metrics using the established concept of “acceptability regions”, which then allows them to score for correctness. Swapping object types, colors, etc in the same scene is easy using the synthetic pipeline and gives an evaluable sense of robustness.

They test a broad set of VLMs, including multilingual ones in 109 translated languages. Some models better represent egocentric vs object centric relations.

Overall, they find that the VLMs do have some acceptable egocentric understanding capabilities but they have severe robustness issues. They find that English notions of frame of reference are held across most test languages, not reflecting the diversity of notions.

**Strengths:**

Deep utilization of synthetic pipeline to really probe spatial perception from several angles.

Demonstrations of anglewise judgements are compelling and surprising! (eg., figure 7)

Great to see the extra effort of testing

Conclusions and discussion are thoughtful and well supported.

**Weaknesses:**

~Sole reliance on synthetic data.~ A brief ablation over natural images would be nice to see to give a sense of how well these findings generalize. They added the case study, the results generalize (though the case study is still close to ID the synth data)!

~Limited diversity of backgrounds. This is another form of variation that may have important implications for generalization.~ They added the case study, the findings generalize!

The multilingual experiments, while welcome, have limited time to breathe in the overall paper with just a few paragraphs and tables stuck in appendices. However, I think this is a great problem to have as the paper is densely packed with quality experimentation.

**Questions:**

Small grammar errors, eg 476: “Does multilingual VLMs..” -> “Do multilingual…” Maybe do a brief grammarly check?

---

> ### Author Response · Authors · 2024-11-25
> **Response to Reviewer QwFz**
>
> Thank you for your thoughtful and constructive feedback and for recognizing our novel approach to probing spatial perception, demonstration of anglewise judgments, and extensive experiments to support the conclusions and discussions. We address your questions below and have updated our paper according to your suggestions.
>
> ### Response to Weakness #1 (synthetic data)
> Thanks for your suggestion. We use synthetic data for the following reasons:
> It’s hard to rotate the target object around the reference object in the real world with uniform steps for collecting natural images.
> We also want to carefully control other variables (e.g., camera pose) in the experiments for systematic analysis, which are hard to maintain in capturing natural images.
> However, we added a case study to see if results in synthetic images generalize to real photos. We fixed the camera pose and manually rotated the target object around the reference object using (1) two equally sized red and blue balls; and (2) a laptop and the red ball. We keep everything else identical to the original COMFORT setup. We found that our findings still hold for alternative backgrounds. Please refer to Section Appendix C for more details.
>
> ### Response to Weakness #2 (limited diversity of backgrounds)
> Thanks for your suggestion. We added a new dataset for a case study with a brown background, while keeping everything else identical to the original COMFORT-BALL setup.
> We found that our findings still hold for alternative backgrounds. Please refer to Section Appendix C for more details.
>
> ### Response to Weakness #3 Limited sections on multilingual experiments
> Thanks for your suggestion. Our multilingual experiments focus on investigating the preferred FoR in different languages. Since we built on the experimental setup already laid out for the English language in prior sections, we focused only on the experiments for multiple languages, which may explain why this section appears more concise. Table 10 in the appendix mainly serves as a full record for readers who are interested in looking at specific languages, but we plotted the preferred FoR on the world map in the main paper.

---

> > ### Comment · Reviewer_QwFz · 2024-12-03
> > **Awesome job with the extra experiment!**
> >
> > I really appreciate the addition of the real image case study (even though it's very small) in Appendix C. This really drives home that this finding generalizes, even though I agree that all your explanations for why synthetic data only is used are valid. Ditto for the limited diversity of backgrounds. I will raise my score as I have no other complaints.

---

> > > ### Author Response · Authors · 2024-12-03
> > >
> > > Thank you for your careful review and thoughtful feedback throughout the process! We’re glad that the addition of the real image case study in Appendix C resonated with you and helped demonstrate the generalizability of our findings. Your active engagement and constructive discussions have been instrumental in improving this work!

---

### Author Response · Authors · 2024-11-25
**General Response to Reviewers and ACs**

We thank all the reviewers and ACs for their time and effort in reviewing the paper and providing valuable and constructive feedback. We appreciate that reviewers have recognized the following contributions of our paper:

### Contributions
- **A novel, systematic, rigorous benchmark to probe and test spatial reasoning**
  - Synthetic pipeline is deeply utilized and angle-wise evaluation is compelling and surprising. *(QwFz)*
  - Continuous definition based on regions of acceptability is an important distinction. *(4ug1)*
  - This novel approach highlights the importance of considering linguistic and cultural variations in model performance. *(GnST)*
  - The authors create challenging but realistic queries using synthetic 3D images and corresponding textual descriptions and propose systematic metrics. *(Jhao)*

- **Experiments are comprehensive with detailed and extensive analysis to support conclusions**
  - The extra effort of testing. Conclusions and discussion are thoughtful and well-supported. *(QwFz)*
  - The experimental setup is quite thorough and shows clear trends, which the authors use to justify their claims. *(4ug1)*
  - The research is thorough and methodical, assessing nine state-of-the-art VLMs with detailed analyses of their robustness, consistency, and flexibility. *(GnST)*
  - The authors tested different VLMs and analyzed how they internally understand spatial representations. The testing methods are reliable. *(Jhao)*

- **Well-motivated work with valuable insights into AI alignment**
  - Motivating statements and scoping within existing work are very clear and compelling. *(4ug1)*
  - The authors extended the test cases to cover 109 languages across 170 regions, further testing how VLMs respond to ambiguities in spatial relationship descriptions under different languages. *(Jhao)*
  - This work broadens the scope by emphasizing ambiguity in spatial reasoning due to different frameworks of reference and cross-linguistic/cross-cultural diversity. The paper’s contributions are highly significant in the context of both vision-language research and broader discussions on AI alignment with human cognition. The cross-lingual evaluation across diverse cultural backgrounds provides valuable insights into the limitations of current VLMs in handling non-English spatial conventions. *(EAzE)*

- **Clear and well-structured writing**
  - The paper's writing is generally clear, and I found the organizational structure helpful for understanding the concepts introduced. *(4ug1)*
  - The paper is well-organized and clearly written, making complex concepts accessible. *(GnST)*
  - The paper is well-structured, with clear explanations of key concepts such as frames of reference and spatial reasoning. *(EAzE)*

### Revisions Made
To address reviewers’ suggestions, we made the following revisions (marked in blue in the PDF):

1. **New experiments for case studies**
   - **2 new case studies of real images** *(COMFORT-BALL, COMFORT-CAR)* for testing if the results generalize to real photos:
     - We fixed the camera pose and manually rotated the target object around the reference object using:
       1. Two equally sized red and blue balls.
       2. A laptop and the red ball.
     - We found that our findings still hold for real images.
   - **1 new case study (COMFORT-BALL)** for testing if the results generalize to synthetic images with a brown background:
     - We added a new dataset for a case study with a brown background while keeping everything else identical to the original COMFORT-BALL setup.
     - We found that our findings still hold for alternative backgrounds.

---

### Meta-Review · Area_Chair_NfCH · 2024-12-22

**Metareview:**

This paper proposes a novel evaluation protocol and benchmark, COnsistent Multilingual Frame Of Reference Test (COMFORT), to examine the spatial reasoning capabilities of vision and language models and evaluates 9 state-of-the-art models using this framework and demonstrate their  significant shortcomings. This is an interesting framework and accompanying analysis of models and reviewers suggest that this paper will be a good contribution to the conference.

**Additional Comments On Reviewer Discussion:**

The rebuttal period resulted in good outcomes and authors have improved their framework based on the reviewer comments and suggestions by adding new case studies.

---

### Decision · Program_Chairs · 2025-01-22

Accept (Oral)